# Analyzing Time-independent Classifiers for Conditional Generation

## Abstract

Classifier guidance diffusion models have advanced conditional image generation by training a **time-dependent** classifier on noisy data from every diffusion timestep to guide denoising process. We revisit this paradigm and show that such dense guidance is unnecessary: a small set of **time-independent** classifiers, trained on data from selected timesteps, suffices to produce high-quality, class-consistent samples. Theoretically, we first analyze the feasibility of using a single time-independent classifier trained on clean data to guide generation under certain conditions which are unrealistic in practice. To address the limitations of real-world image data, we then extend this approach to a small set of classifiers trained on noisy data from some timesteps and derive a convergence bound that depends on the number of classifiers employed. Experiments on both synthetic and real-world datasets demonstrate that guiding an unconditional diffusion model with only a few time-independent classifiers achieves performance comparable to models guided by a fully time-dependent classifier.

## 1 Introduction

In recent years, denoising diffusion probabilistic models (DDPMs) (Ho et al., 2020; Nichol & Dhariwal, 2021; Sohl-Dickstein et al., 2015; Song et al., 2020) have emerged as powerful generative models capable of producing data of quality comparable to that of GANs (Brock et al., 2018; Goodfellow et al., 2014; Karras et al., 2019), spanning modalities such as images (Zhang et al., 2023; Ho & Salimans, 2022; Dhariwal & Nichol, 2021; Ramesh et al., 2021), videos (Ho et al., 2022b;a), and audio (Kong et al., 2020). A DDPM consists of a forward process that gradually perturbs clean training data by increasing the noise scale, and a reverse process that reconstructs the original data distribution. As a result, DDPMs can generate high-quality novel samples by initiating the reverse process from standard Gaussian noise (Ho et al., 2020).

Conditional generation (Song et al., 2021; Dhariwal & Nichol, 2021; Ho & Salimans, 2022) is a key problem in DDPMs, enabling condition-consistent sample generation such as class-specific images. A representative approach is the classifier-guided diffusion model (CGDM) (Dhariwal & Nichol, 2021), which uses a **time-dependent** classifier to guide the generation process. Specifically, Song et al. (2021) proposed constructing intermediate conditional distributions $p_t(\boldsymbol{x} \mid y)$ in the reverse process using conditional score functions, so that it can finally generate the target conditional distribution $p_0(\boldsymbol{x} \mid y)$. CGDM employs this idea by decomposing the conditional score function into an unconditional score function together with a guidance term provided by a time-dependent classifier $p_t(y \mid \boldsymbol{x})$. Although this strategy enables high-quality conditional generation, it requires training the classifier on noisy data at every timestep of the forward process, which is computationally expensive and labor-intensive.

In this paper, we investigate whether training a single **time-independent** classifier on clean data, or a small set of time-independent classifiers on noisy data from a few timesteps, can still provide sufficient guidance for conditional generation. Our key observation is that the target conditional distribution can be expressed as $p_0(\boldsymbol{x} \mid y) \propto p_0(\boldsymbol{x})p_0(y \mid \boldsymbol{x})$, which suggests that it suffices to generate intermediate distributions of the form $p_t(\boldsymbol{x})p_0(y \mid \boldsymbol{x})$ in the reverse process, rather than the full conditional distributions $p_t(\boldsymbol{x} \mid y) \propto p_t(\boldsymbol{x})p_t(y \mid \boldsymbol{x})$. To achieve this, under DDPM framework in discrete settings, we construct a transition probability and show that it can guide the reverse process generating $\boldsymbol{x}_{t-1} \sim p_{t-1}(\boldsymbol{x}_{t-1})p_0(y \mid \boldsymbol{x}_{t-1})$ if $\boldsymbol{x}_t \sim p_t(\boldsymbol{x}_t)p_0(y \mid \boldsymbol{x}_t)$. Crucially,

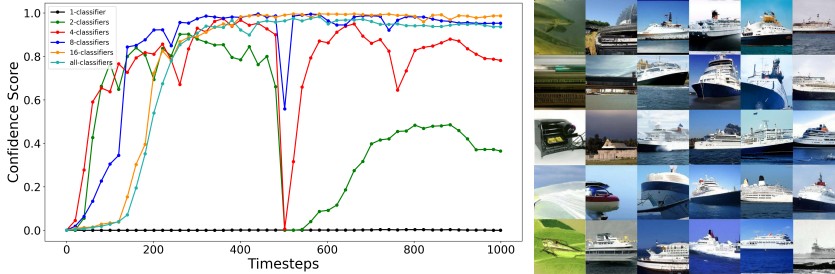

Figure 1: The confidence scores of classifiers during the reverse diffusion process and the generated images of $256 \times 256$ resolution by an unconditional diffusion model are guided by classifiers of (1, 2, 4, 8, 16, all) timesteps, where all is 1000, with each classifier represented by a different color. The class is "ocean liner".

this transition probability depends only on the time-independent classifier $p_0(y \mid \boldsymbol{x})$, making it the single classifier required to generate the target conditional distribution.

However, the above analysis requires sampling $\boldsymbol{x}_T \sim p_T(\boldsymbol{x}_T)p_0(y \mid \boldsymbol{x}_T)$ for initializing the reverse process, which is intractable. Although $p_T(\boldsymbol{x}) \approx \mathcal{N}(\boldsymbol{x}; 0, I)$ for large $T$, the complexity of the classifier $p_0(y \mid \boldsymbol{x})$ makes it impossible to sample directly from $\mathcal{N}(\boldsymbol{x}; 0, I)p_0(y \mid \boldsymbol{x})$. This raises the question of whether we can instead sample $\boldsymbol{x}_T \sim \mathcal{N}(\boldsymbol{x}_T; 0, I)$. Under a mild smoothness assumption on the unconditional score function, we prove that if the classifier $p_0(y \mid \boldsymbol{x})$ is strongly log-concave, then starting from $\mathcal{N}(\boldsymbol{x}; 0, I)$ can still converge to the target conditional distribution exponentially in $T$. Furthermore, experiments on both synthetic and real-world image datasets corroborate this result.

In practice, an additional challenge arises. The above analysis requires $p_0(y \mid \cdot)$ to be defined on the entire space $\mathbb{R}^n$ so that it can provide guidance even for noisy data. However, this condition does not hold in real-world scenarios due to the manifold hypothesis (Bengio et al., 2013), which states that real-world data typically lie on a low-dimensional submanifold $\mathcal{M}_0 \subset \mathbb{R}^n$. As a result, $p_0(y \mid \cdot)$ is only meaningful in a neighborhood of the data manifold $\mathcal{M}_0$ and fails to provide informative guidance for noisy data far from $\mathcal{M}_0$. To address this limitation, we propose training a small number of time-independent classifiers on noisy data at selected timesteps so that they remain informative for noisy inputs. Theoretically, we show that the total variation distance between the distribution generated by our model guided by $k$ classifiers and the target conditional distribution is bounded by $\mathcal{O}(1/k)$. In practice, $k$ can be chosen much smaller than the number of diffusion timesteps $T$. For example, experiments on ImageNet-1K (Deng et al., 2009) demonstrate that with only $k = 8$ classifiers, the reverse process still produces high-quality samples, and evaluation metrics such as FID and sFID remain comparable to those achieved by CGDM with $T = 1000$ classifiers.

In conclusion, our contributions include the following three aspects.

(i) We theoretically prove that only using the time-independent classifier $p_0(y \mid \boldsymbol{x})$ trained on clean data can also guide the reverse process to generate the conditional distribution if we can sample $\boldsymbol{x}_T \sim p_T(\boldsymbol{x}_T)p_0(y \mid \boldsymbol{x}_T)$, and also our synthetic experiment shows the validity of this result.

(ii) To relax the initialization requirement, we analyze the possibility of drawing the initial sampling from $\mathcal{N}(\boldsymbol{x}; 0, I)$. Under a smoothness assumption of unconditional score function, theoretical result shows that if $p_0(y \mid \cdot)$ is strongly log-concave, then initialization from $\mathcal{N}(\boldsymbol{x}; 0, I)$ still ensures reliable generation. Experiments on both synthetic and real-world datasets confirm this result.

(iii) To deal with real-world image datasets, because of the manifold hypothesis, we propose to train a few time-independent classifiers on noisy data of some timesteps to guide the generation. Theoretically, we derive an upper bound on the total variation distance between the generated distribution and the target distribution in terms of the number of classifiers. Empirically, experiments on ImageNet-1K show that even with a small number of classifiers, our method achieves competitive performance.

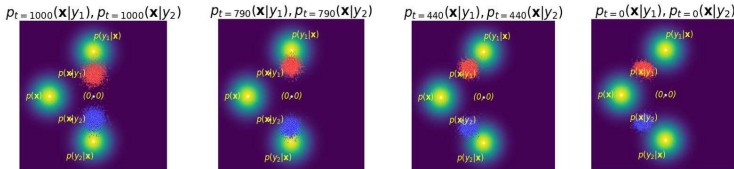

Figure 2: For $i = 1, 2$, reverse diffusion process that initially samples from the distributions $\mathcal{N}(\boldsymbol{x}; 0, I)p_0(y_i|\boldsymbol{x})$, and the classifier guided diffusion model reconstructs the conditional distribution $p_0(\boldsymbol{x}|y = y_i)$.

## 2 RELATED WORKS

**Diffusion Model.** Denoising diffusion probabilistic models (DDPMs) (Sohl-Dickstein et al., 2015; Ho et al., 2020) have become a powerful paradigm for data generation (Zhang et al., 2023; Rombach et al., 2022). Score-based generative models (Song & Ermon, 2019) estimate data gradients and sample with Langevin dynamics, and Song et al. (2021) unified them with DDPMs via stochastic differential equations. While DDPMs produce high-quality images, they require many steps, leading to high computational costs. To improve efficiency, Song et al. (2020) proposed denoising diffusion implicit models (DDIMs), which generalize DDPMs with non-Markovian processes while preserving the same training objective. DDIMs achieve comparable quality with fewer steps. Building on these foundations, recent works (Zhang et al., 2023; Ho & Salimans, 2022; Dhariwal & Nichol, 2021; Rombach et al., 2022; Peebles & Xie, 2023; Stypułkowski et al., 2024; Tevet et al., 2022; Ramesh et al., 2022) have demonstrated the broad applicability of diffusion models across diverse domains.

In theoretical analysis, many works research the distance between the target distribution and the generated distribution. These works study the convergence guarantees for ODE-based samplers(Huang et al., 2024; Chen et al., 2023b) and SDE-based samplers (Bortoli et al., 2021; Li & Yan, 2024). Moreover, they propose many techniques for relaxing the assumptions. Especially, in order to weaken the assumptions of smoothness, the technique of applying Girsanov's theorem (Chen et al., 2023c) has been proposed for analyzing SDE-based samplers.

**Conditional Generation.** Conditional generation is a key task in diffusion models. Dhariwal & Nichol (2021) introduced the classifier-guided diffusion model (CGDM), which uses an auxiliary time-dependet classifier to guide the reverse process and improves class-conditional sampling quality. However, the classifier must be trained on noisy samples from every timestep, which is costly. Our work shows that this is unnecessary: classifiers from only a few timesteps—or even timestep zero—are sufficient for conditional generation.

Ho & Salimans (2022) proposed classifier-free guidance, which removes the classifier by approximating classifier gradients with score function differences, but still requires labeled data for training. Other approaches, such as off-the-shelf and plug-and-play methods (Ma et al., 2023; Go et al., 2023; Graikos et al., 2022; Nguyen et al., 2017; Chao et al., 2022; Huang et al., 2022), reduce classifier training by reusing pretrained models, though their focus lies beyond the scope of our work.

## 3 INVESTIGATION OF THE CLASSIFIER GUIDANCE

### 3.1 PRELIMINARIES

**Diffusion model.** Diffusion model (Ho et al., 2020) is a method for generating new samples $\boldsymbol{x} \sim p(\boldsymbol{x})$. It first draws sample $\boldsymbol{x}_0 \sim p_0(\boldsymbol{x}_0) = p(\boldsymbol{x}_0)$ and then gradually add noise to $\boldsymbol{x}_0$ so that after sufficient steps the $\boldsymbol{x}_T$ approximately obeys $\mathcal{N}(\boldsymbol{x}_T; 0, I)$. Formally, $\boldsymbol{x}_t \sim p_t(\boldsymbol{x}_t)$ is given by $p(\boldsymbol{x}_t \mid \boldsymbol{x}_{t-1}) = \mathcal{N}(\boldsymbol{x}_t; \sqrt{1 - \beta_t}\boldsymbol{x}_{t-1}, \beta_t I)$. Next, the denoising process is to gradually generate clean samples by learning $p(\boldsymbol{x}_{t-1} \mid \boldsymbol{x}_t) = \mathcal{N}(\boldsymbol{x}_{t-1}; \boldsymbol{\mu}_t(\boldsymbol{x}_t), \sigma_t^2 I)$, where $\boldsymbol{\mu}_t(\boldsymbol{x}_t)$ is obtained by removing the noise $\boldsymbol{\epsilon}_t(\boldsymbol{x}_t)$ from $\boldsymbol{x}_t$, i.e., $\boldsymbol{\mu}_t(\boldsymbol{x}_t) = \frac{1}{\sqrt{1-\beta_t}}(\boldsymbol{x}_t - \frac{\beta_t}{\sqrt{1-\bar{\alpha}_t}}\boldsymbol{\epsilon}_t(\boldsymbol{x}_t)), \bar{\alpha}_t := \prod_{s=1}^{t}(1-\beta_s)$. So after $T$ steps of denoising, $\boldsymbol{x}_0$ can be recovered. These two processes can also be expressed by using stochastic differential equation (SDE) framework (Song et al., 2021); see more details in Appendix B.

Figure 3: Generated images of $256 \times 256$ resolutions under the guidance of the different number of classifiers. Left: no classifiers guidance (FID 26.21), middle: 8 classifiers guidance (FID 12.90), right: 1000 classifier guidance (FID 12.00). The ground truth labels are "Maltese dog", "monarch butterfly", "balloon", and "cheeseburger".

**Classifier guidance.** Classifier guidance diffusion models (CGDM) (Dhariwal & Nichol, 2021) generate samples $\boldsymbol{x} \sim p_0(\boldsymbol{x} \mid y)$ by using an additional classifier to guide the reverse process. Specifically, in CGDM, the goal is to generate all intermediate conditional distribution $p_t(\boldsymbol{x} \mid y)$. Since $p_{t-1}(\boldsymbol{x}_{t-1} \mid y) = \int p_t(\boldsymbol{x}_t \mid y) p(\boldsymbol{x}_{t-1} \mid \boldsymbol{x}_t, y) d\boldsymbol{x}_t$, it needs to obtain the transition probability $p(\boldsymbol{x}_{t-1} \mid \boldsymbol{x}_t, y)$,

$$p(\boldsymbol{x}_{t-1} \mid \boldsymbol{x}_t, y) = \frac{p(\boldsymbol{x}_{t-1} \mid \boldsymbol{x}_t)p(y \mid \boldsymbol{x}_{t-1}, \boldsymbol{x}_t)}{p_t(y \mid \boldsymbol{x}_t)} = p(\boldsymbol{x}_{t-1} \mid \boldsymbol{x}_t)\frac{p_{t-1}(y \mid \boldsymbol{x}_{t-1})}{p_t(y \mid \boldsymbol{x}_t)},$$

where we use the fact that $p(y \mid \boldsymbol{x}_{t-1}, \boldsymbol{x}_t) = p_{t-1}(y \mid \boldsymbol{x}_{t-1})$ (Dhariwal & Nichol, 2021). SThis transition consists of two terms: the unconditional transition $p(\boldsymbol{x}_{t-1} \mid \boldsymbol{x}_t)$ and the ratio $p_{t-1}(y \mid \boldsymbol{x}_{t-1})/p_t(y \mid \boldsymbol{x}_t)$, which introduces the new term $\nabla_{\boldsymbol{x}} \log p_t(y \mid \boldsymbol{x}_t)$ in the reverse process. Consequently, the reverse process is

$$\boldsymbol{x}_{t-1} = \boldsymbol{\mu}_t(\boldsymbol{x}_t) + \sigma_t^2 \nabla_{\boldsymbol{x}} \log p_t(y \mid \boldsymbol{x}_t) + \sigma_t \boldsymbol{\varepsilon}, \ \boldsymbol{\varepsilon} \sim \mathcal{N}(0, I), \tag{1}$$

where the classifier guidance $\nabla_{\boldsymbol{x}} \log p_t(y \mid \boldsymbol{x})$ is time-dependent and requires training classifiers on noisy data for all timestep $t$.

## 3.2 TIME-INDEPENDENT CLASSIFIER GUIDANCE

Let us reconsider the reverse process in CGDM. The main technique is applying the transition probability $p(\boldsymbol{x}_{t-1} \mid \boldsymbol{x}_t, y)$ that can generate $\boldsymbol{x}_{t-1} \sim p_{t-1}(\boldsymbol{x}_{t-1} \mid y)$ from $\boldsymbol{x}_t \sim p_t(\boldsymbol{x}_t \mid y)$ step by step, ultimately yielding $p_0(\boldsymbol{x} \mid y)$. However, the goal is to generate $p_0(\boldsymbol{x} \mid y)$, instead of all $p_t(\boldsymbol{x} \mid y)$. Noting that $p_0(\boldsymbol{x} \mid y) \propto p_0(\boldsymbol{x})p_0(y \mid \boldsymbol{x})$, we observe that it suffices to generate $\boldsymbol{x}_{t-1} \sim p_{t-1}(\boldsymbol{x}_{t-1})p_0(y \mid \boldsymbol{x}_{t-1})$ from $\boldsymbol{x}_t \sim p_t(\boldsymbol{x}_t)p_0(y \mid \boldsymbol{x}_t)$, then it can complete the goal of generating $p_0(\boldsymbol{x} \mid y)$. This perspective motivates the construction of a new transition probability $\tilde{p}(\boldsymbol{x}_{t-1} \mid \boldsymbol{x}_t, y)$. The following theorem provides a general framework for constructing such a transition probability; see Appendix A.1 for the proof.

**Theorem 3.1.** *For a fixed classifier $h_y(\boldsymbol{x})$, if we draw $\boldsymbol{x}_t \sim Z_t p_t(\boldsymbol{x}_t) h_y(\boldsymbol{x}_t)$ and generate $\boldsymbol{x}_{t-1}$ by applying the transition probability*

$$\tilde{p}(\boldsymbol{x}_{t-1} \mid \boldsymbol{x}_t, y) := p(\boldsymbol{x}_{t-1} \mid \boldsymbol{x}_t)\frac{h_y(\boldsymbol{x}_{t-1})}{h_y(\boldsymbol{x}_t)},$$

*then the generated*

$$\boldsymbol{x}_{t-1} \sim Z_{t-1} p_{t-1}(\boldsymbol{x}_{t-1}) h_y(\boldsymbol{x}_{t-1}),$$

*where $Z_t$ and $Z_{t-1}$ are normalization terms.*

Therefore, if we set the classifier $h_y := p_0(y \mid \cdot)$ in Theorem 3.1 and draw

$$\boldsymbol{x}_T \sim Z_T p_T(\boldsymbol{x}_T) p_0(y \mid \boldsymbol{x}_T) \ \Rightarrow \ \boldsymbol{x}_{T-1} \sim Z_{T-1} p_{T-1}(\boldsymbol{x}_{T-1}) p_0(y \mid \boldsymbol{x}_{T-1}),$$

$$\Rightarrow \cdots,$$

$$\Rightarrow \ \boldsymbol{x}_0 \sim Z_0 p_0(\boldsymbol{x}_0) p_0(y \mid \boldsymbol{x}_0) = p_0(\boldsymbol{x}_0 \mid y),$$

---

**Algorithm 1** Sampling using diffusion $(\boldsymbol{\mu}_\theta(\boldsymbol{x}_t), \Sigma_\theta(\boldsymbol{x}_t))$ and $k$ classifiers $p_{t_i}(y \mid \boldsymbol{x}_t)$

---

**Input:** class label $y$, gradient scale $s$
$\boldsymbol{x}_T \leftarrow$ sample from $N(\boldsymbol{x}_T; 0, I)$ and compute time interval $t^* \leftarrow T/k$
**for all** $t$ from $T$ to $1$ **do**
$\quad i \leftarrow \lfloor t/t^* \rfloor$
$\quad \boldsymbol{\mu}, \Sigma \leftarrow \boldsymbol{\mu}_\theta(\boldsymbol{x}_t), \Sigma_\theta(\boldsymbol{x}_t)$
$\quad \boldsymbol{x}_{t-1} \leftarrow \mathcal{N}(\boldsymbol{\mu} + s\Sigma\nabla_{\boldsymbol{x}_t} \log p_i(y \mid \boldsymbol{x}_t), \Sigma)$
**end for**
**return** $\boldsymbol{x}_0$

---

then we can successfully recover $p_0(\boldsymbol{x} \mid y)$. The intermediate $\boldsymbol{x}_t$ just needs to obey $p_t(\boldsymbol{x}_t)p_0(y \mid \boldsymbol{x}_t)$ for all $t$, which can be achieved by applying the transition probability $\tilde{p}(\boldsymbol{x}_{t-1} \mid \boldsymbol{x}_t, y)$. Importantly, this transition probability requires only the knowledge of the single time-independent classifier $p_0(y \mid \cdot)$. More specifically, the reverse process is characterized by the following proposition and the proof is provided in Appendix A.2.

**Proposition 3.2.** *Using the same notations as in Theorem 3.1, if $\boldsymbol{x}_t \sim Z_t p_t(\boldsymbol{x}_t) h_y(\boldsymbol{x}_t)$ and generating $\boldsymbol{x}_{t-1}$ by*

$$\boldsymbol{x}_{t-1} = \boldsymbol{\mu}(\boldsymbol{x}_t) + \sigma_t^2 \nabla_{\boldsymbol{x}} \log h_y(\boldsymbol{x}_t) + \sigma_t \boldsymbol{\varepsilon}, \; \boldsymbol{\varepsilon} \sim \mathcal{N}(0, I), \tag{2}$$

*then $\boldsymbol{x}_{t-1} \sim Z_{t-1} p_{t-1}(\boldsymbol{x}_{t-1}) h_y(\boldsymbol{x}_{t-1})$, where $Z_t$ and $Z_{t-1}$ are normalization terms.*

By comparing the reverse dynamics in (2) and (1), we can see that the classifier guidance term $\nabla_{\boldsymbol{x}} \log p_0(y \mid \boldsymbol{x})$ in our model is time-independent and can be trained solely on clean data. We construct a synthetic experiment to test the validity of the reverse dynamics (2); see the results in Section 4.1.

## 3.3 INITIAL SAMPLING AND CONTRACTIVE PROPERTY

The next challenge is handling the initial sampling $\boldsymbol{x}_T \sim p_T(\boldsymbol{x}_T)p_0(y \mid \boldsymbol{x}_T)$. Although $p_T(\boldsymbol{x})$ approximates $\mathcal{N}(\boldsymbol{x}; 0, I)$, the complexity of $p_0(y \mid \boldsymbol{x})$ results in sampling from $\mathcal{N}(\boldsymbol{x}; 0, I)p_0(y \mid \boldsymbol{x})$ intractable. This raises the question of whether we can instead directly sample $\boldsymbol{x}_T \sim \mathcal{N}(\boldsymbol{x}_T; 0, I)$. To answer this question, we analyze the contractive property of the reverse dynamics (2) in the following theorem. A more formal statement of Theorem 3.3, along with its proof, is provided in Appendix B.1.

**Theorem 3.3** (Informal). *Under a mild smoothness assumption on the unconditional $\log p_t$, if the time-independent classifier $h_y$ is $M$-strongly log-concave, i.e., $-\nabla_{\boldsymbol{x}}^2 \log h_y(\boldsymbol{x}) \succeq MI$ for some constant $M > 0$, then even when $\boldsymbol{x}_T \sim \mathcal{N}(\boldsymbol{x}_T; 0, I)$ is used as the initialization in the reverse process (2), the generated distribution converges to the target distribution exponentially in $T$.*

The main idea of Theorem 3.3 is to establish a contractive inequality for (2), given by

$$\|\bar{\boldsymbol{x}}_t - \hat{\boldsymbol{x}}_t\|^2 \leq e^{-K(T-t)}\|\bar{\boldsymbol{x}}_T - \hat{\boldsymbol{x}}_T\|^2, \tag{3}$$

where $\bar{\boldsymbol{x}}_t$ and $\hat{\boldsymbol{x}}_t$ are generated by (2) from different initializations $\bar{\boldsymbol{x}}_T$ and $\hat{\boldsymbol{x}}_T$, respectively. Owing to the smoothness of $\log p_t$ and the strong log-concavity of $p_0(y \mid \cdot)$, we can ensure the existence of a positive constant $K > 0$. Inequality (3) then implies that the distance between $\bar{\boldsymbol{x}}_t$ and $\hat{\boldsymbol{x}}_t$ decays exponentially in $T$. Consequently, sampling $\boldsymbol{x}_T \sim \mathcal{N}(\boldsymbol{x}_T; 0, I)$ has little impact on the final generation compared with $\boldsymbol{x}_T \sim p_T(\boldsymbol{x}_T)p_0(y \mid \boldsymbol{x}_T)$.

For the strong log-concavity of the classifier $h_y$, a simple example is the "Gaussian-like" classifier of the form $h_y(\boldsymbol{x}) = \exp(-\|\boldsymbol{x} - \boldsymbol{\mu}_y\|^2/\sigma_y^2)$, which has been used in the noise inverse problem (Dhariwal & Nichol, 2021). It is clear that such an $h_y$ is $1/\sigma_y^2$-strongly log-concave. Using this type of classifier, we construct synthetic datasets to verify the results of Theorem 3.3; see Appendix D.1 for details. For real-world image datasets, although the strong log-concavity of the classifier cannot be guaranteed, our experiments demonstrate that initialization with $\mathcal{N}(\boldsymbol{x}; 0, I)$ remains valid; see Section 4.2. We also empirically evaluate the contractive property on real-world image datasets, and the results are shown in Section 4.3.

### 3.4 Manifold Hypothesis and More Classifiers

In previous analysis, we omitted an important assumption that $p_0(y \mid \cdot)$ is meaningful even for noisy data $\boldsymbol{x}_t$ at large $t$. However, this assumption is generally not satisfied for real-world image datasets. High-dimensional data typically concentrate on a much lower-dimensional submanifold, a phenomenon known as the manifold hypothesis (Bengio et al., 2013), which has been extensively examined in both theory (Fefferman et al., 2016) and experiments (Brown et al., 2022). Since clean data in $\mathbb{R}^n$ lie on a low-dimensional submanifold $\mathcal{M}_0 \subset \mathbb{R}^n$, a classifier $p_0(y \mid \cdot)$ trained on clean data is only meaningful within a small neighborhood of $\mathcal{M}_0$. Consequently, $p_0(y \mid \cdot)$ cannot provide reliable guidance for noisy samples $\boldsymbol{x}_t$ far from the data manifold $\mathcal{M}_0$.

Our next goal is to address this limitation. Motivated by the approach of Song & Ermon (2019), we train a small number of time-independent classifiers on noisy data from different timesteps of the forward diffusion process so that they remain informative even for noisy inputs. Let $T = t_0 > t_1 > \cdots > t_k = 0$. For each $t_i$, we train a classifier $h_y^i := p_{t_i}(y \mid \cdot)$ on noisy data $\boldsymbol{x}_{t_i}$, for $i = 1, 2, \ldots, k$. During generation, from step $t_{i-1}$ to step $t_i$, we employ $h_y^i$ to guide sampling according to the reverse dynamics (2).

However, two additional issues must be addressed: how to design the transition probability at each step $t_i$ when the classifier changes, and how to modify the reverse dynamics (2) at $t_i$. First, for $i = 1, 2, \ldots, k - 1$, the transition probability at $t_i$ can be defined as

$$\tilde{p}_{t_i}(\boldsymbol{x}_{t_i} \mid \boldsymbol{x}_{t_{i+1}}, y) := p(\boldsymbol{x}_{t_i} \mid \boldsymbol{x}_{t_{i+1}}) \frac{h_y^{i+1}(\boldsymbol{x}_{t_i})}{h_y^i(\boldsymbol{x}_{t_{i+1}})} = p(\boldsymbol{x}_{t_i} \mid \boldsymbol{x}_{t_{i+1}}) \frac{p_{t_{i+1}}(y \mid \boldsymbol{x}_{t_i})}{p_{t_i}(y \mid \boldsymbol{x}_{t_{i+1}})}. \tag{4}$$

Under this transition probability, when $\boldsymbol{x}_{t_{i+1}} \sim p_{t_{i+1}}(\boldsymbol{x}_{t_{i+1}}) p_{t_i}(y \mid \boldsymbol{x}_{t_{i+1}})$, it follows that $\boldsymbol{x}_{t_i} \sim p_{t_i}(\boldsymbol{x}_{t_i}) p_{t_{i+1}}(y \mid \boldsymbol{x}_{t_i})$. The reasoning is analogous to the proof of Theorem 3.1; see Appendix A.1. Based on this transition probability, the reverse dynamics is formulated in the following proposition, with proof provided in the Appendix A.3.

**Proposition 3.4.** *Using the same notations as the above, the desired $\boldsymbol{x}_{t_i}$ can be generated from $\boldsymbol{x}_{t_{i+1}}$ by*

$$\boldsymbol{x}_{t_i} = \boldsymbol{\mu}(\boldsymbol{x}_{t_{i+1}}) + \sigma_{t_{i+1}}^2 \nabla_{\boldsymbol{x}} \log p_{t_i}(y \mid \boldsymbol{x}_{t_{i+1}}) + \sigma_{t_{i+1}} \boldsymbol{\varepsilon}.$$

Except at the timesteps $t_i$ where the guidance changes, the transition from $\boldsymbol{x}_{t_{i-1}}$ to generate $\boldsymbol{x}_{t_{i+1}}$ follows the same reverse dynamics as in (2), guided by $p_{t_{i+1}}(y \mid \boldsymbol{x})$. In other words, the term $\nabla_{\boldsymbol{x}} \log p_{t_i}(y \mid \boldsymbol{x})$ is used from step $t_{i-1} - 1$ to step $t_i$, while the term $\nabla_{\boldsymbol{x}} \log p_{t_{i+1}}(y \mid \boldsymbol{x})$ is used from step $t_i - 1$ to step $t_{i+1}$. The complete sampling pipeline is summarized in Algorithm 1, where we also introduce a guidance scale $s$ to control the strength of guidance.

The next question is how to determine the number $k$ of classifiers. To this end, we investigate whether it is possible to establish an upper bound on the distance between the distribution generated with $k$ classifiers and the target conditional distribution. To address this, we apply Girsanov's theorem (Liptser & Shiryaev, 2013) to bound the total variation between the target distribution $p_y := p(\cdot \mid y)$ and the generated distribution $\tilde{p}_y$. A formal statement of the following theorem and its proof are provided in Appendix B.2.

**Theorem 3.5** (Informal). *Under some assumptions, we have that*

$$\mathsf{TV}(p_y, \tilde{p}_y) \leq \mathcal{O}\left(\frac{1}{k}\right).$$

Here, we outline the key ideas in the proof of this theorem, which consists of two parts. First, Girsanov's theorem is applied to relate the total variation (TV) to the difference in guidance terms, i.e.,

$$\mathsf{TV}(p_y, \tilde{p}_y) \leq \sum_{i=1}^{k} \int_{t_i}^{t_{i-1}} \mathbb{E}\left[\|p_{t_i}(y \mid \boldsymbol{x}_t) - p_t(y \mid \boldsymbol{x}_t)\|^2\right] dt,$$

an idea inspired by Bortoli et al. (2021); Chen et al. (2023c). However, we employ another proof without considering the Wiener space as the previous works did. Second, to obtain an upper bound for the term on the right-hand side, we apply Grönwall's Inequality under suitable assumptions on the target conditional distribution.

Table 1: Comparison between the different numbers of classifier guidance on sample quality.

| SIZE | CONDITIONAL | CLASSIFIERS | FID | SFID |
|------|-------------|-------------|-----|------|
| 256 | ✗ | 0 | 26.21 | 6.35 |
| 256 | ✗ | 4 | 14.81 | 8.51 |
| 256 | ✗ | 8 | 12.90 | 11.09 |
| 256 | ✗ | 16 | 12.33 | 11.43 |
| 256 | ✗ | 1000 | 12.00 | 10.40 |
| 256 | ✓ | 0 | 10.94 | 6.02 |
| 256 | ✓ | 8 | 4.78 | 5.22 |
| 256 | ✓ | 1000 | 4.59 | 5.25 |
| 128 | ✓ | 0 | 5.91 | 5.09 |
| 128 | ✓ | 8 | 3.05 | 5.18 |
| 128 | ✓ | 1000 | 2.97 | 5.09 |
| 64 | ✓ | 8 | 4.79 | 6.07 |
| 64 | ✓ | 1000 | 4.14 | 5.82 |

Experimentally, we investigate the confidence score of the generated results guided by different number of time-independent classifiers provided by Nichol & Dhariwal (2021). As shown in Figure 1, the model fails to provide meaningful guidance when only a single classifier at timestep 0 is used. This is because one classifier cannot provide reliable guidance for noisy data far from the data manifold $\mathcal{M}_0$, as discussed earlier. In contrast, when the number of classifiers increases to $k = 8$ or $k = 16$, the reverse process produces results with performance comparable to CGDM, which relies on all 1000 classifiers. These findings empirically validate our approach.

## 4 EXPERIMENTAL RESULTS

In this section, we present experiments on both synthetic and real-world datasets to validate the proposed theory, with implementation details provided in Appendix C.

### 4.1 ONE CLASSIFIER GUIDANCE FOR SYNTHETIC DATA

As discussed in Theorem 3.1, one classifier can be sufficient for conditional generation guidance, if the initial sampling condition can be satisfied and the classifier is meaningful even for noisy data. In this subsection, we experiment on 2-dimensional toy datasets to verify this.

Let the clean data be drawn from $p_0(\boldsymbol{x}) = \mathcal{N}(\boldsymbol{x}; \boldsymbol{\mu}_0, \Sigma)$. Suppose it has two classes $y \in \{y_1, y_2\}$ with classifiers set by $p_0(y = y_1 \mid \boldsymbol{x}) = \mathcal{N}(\boldsymbol{x}; \boldsymbol{\mu}_1, \Sigma)$ and $p_0(y = y_2 \mid \boldsymbol{x}) = \mathcal{N}(\boldsymbol{x}; \boldsymbol{\mu}_2, \Sigma)$, where $\Sigma = I$, $\boldsymbol{\mu}_0 = (-6, 0)$, $\boldsymbol{\mu}_1 = (0, 6)$, and $\boldsymbol{\mu}_2 = (0, -6)$. Under these settings, by $p_0(\boldsymbol{x} \mid y) \propto p_0(\boldsymbol{x}) \cdot p_0(y \mid \boldsymbol{x})$, the objective conditional distributions $p_0(\boldsymbol{x} \mid y = y_1)$ and $p_0(\boldsymbol{x} \mid y = y_2)$ are also normal distributions with expectations $\boldsymbol{\mu}_{\boldsymbol{x}_0|y_1} = (\boldsymbol{\mu}_0 + \boldsymbol{\mu}_1)/2$ and $\boldsymbol{\mu}_{\boldsymbol{x}_0|y_2} = (\boldsymbol{\mu}_0 + \boldsymbol{\mu}_2)/2$ and variances $\Sigma_{\boldsymbol{x}_0|y_1} = \Sigma_{\boldsymbol{x}_0|y_2} = \Sigma/2$.

We randomly sampled 30k two-dimensional points from $p_0(\boldsymbol{x})$ as training data to train an unconditional diffusion model with $T = 1000$. For the reverse process, we sampled 2k two-dimensional points for each class $y_i$ from the distribution $\mathcal{N}(\boldsymbol{x}; 0, I)p_0(y_i \mid \boldsymbol{x})$, for $i = 1, 2$. This sampling is tractable because they are also normal distributions. The final generation results are shown in Figure 2. Both groups of data run the reverse diffusion process guided by their corresponding fixed classifiers $p_0(y_i \mid \boldsymbol{x})$ and successfully reconstruct the distributions $p_0(\boldsymbol{x}_0 \mid y = y_i)$. This verifies Theorem 3.1, demonstrating that the information provided by the classifier $p_0(y \mid \boldsymbol{x})$ alone is sufficient to generate the final conditional distribution.

### 4.2 QUANTITATIVE COMPARISON

Theorem 3.1 shows that guided diffusion does not require a time-dependent classifier trained on noisy data from all timesteps. For real-world image datasets, as discussed in Section 3.4, a small set of time-independent classifiers trained on noisy data from a few timesteps (e.g., 8 timesteps instead

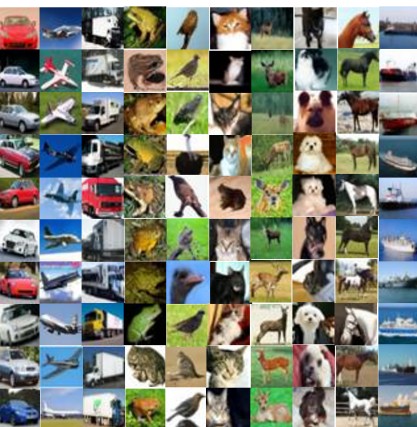

Figure 4: Classifier guided samples on CIFAR-10, each column corresponds to different classes. The upper 5 rows are guided by 1000 classifiers, and the lower 5 rows are guided by 10 classifiers.

Table 2: Samples quality guided by 10 and 1000 classifiers on CIFAR10 dataset.

| CLASSIFIERS | TRAINING ITER | FID | SFID |
|---|---|---|---|
| 1000 | 100K | 19.36 | 18.22 |
| 10 | 30K | 7.36 | 6.91 |

of all 1000) can also guide the diffusion model to reconstruct $p_0(\boldsymbol{x} \mid y)$. In this subsection, we conduct experiments on ImageNet-1K to quantitatively verify this idea. For simplicity, we refer to the use of time-independent classifiers trained on noisy data from $k$ timesteps as "using $k$ classifiers."

We report experimental results using a diffusion model (Nichol & Dhariwal, 2021) trained on ImageNet-1K, with classifiers (Nichol & Dhariwal, 2021) trained on $k = 8$ different timesteps. The total number of timesteps is set to $T = 1000$, and the reverse diffusion process is executed according to Algorithm 1 to guide an unconditional diffusion model in sample generation.

Figure 3 presents images generated by an unconditional diffusion model with the guidance of 0, 8, and 1000 classifiers. The leftmost images show that using 0 classifiers yields poor class consistency, as the diffusion model generates samples without guidance. In contrast, when following the strategy of Algorithm 1, class consistency improves substantially with classifier guidance. The middle images in Figure 3, generated with 8 classifiers, demonstrate that even this small number of classifiers is sufficient to produce high-quality, class-consistent samples. Moreover, the visual quality and detail of these samples are comparable to those generated with 1000 classifiers, i.e., using a time-dependent classifier trained on noisy data from all timesteps. More experiments with larger figure are provided in Appendix E.

To quantitatively demonstrate the performance, we evaluate multiple metrics, including Fréchet Inception Distance (FID) (Heusel et al., 2017), sliding FID, Inception Score (IS) (Szegedy et al., 2016), recall, and precision (Kynkäänniemi et al., 2019). As shown in the top rows of Table 1, the unconditional diffusion model without classifier guidance fails to generate high-quality samples, whereas models with classifier guidance achieve substantial improvements. Guidance with as few as 4 classifiers leads to significant gains across all metrics, and using 8 classifiers yields performance comparable to guidance with all 1000 classifiers. Increasing the number of classifiers to 16 results in only marginal improvements, with performance effectively saturated at the level of 1000 classifiers. Experiments with conditional diffusion report similar results across all image resolutions. These findings validate our theory that only a small number of classifiers are sufficient for effective conditional generation.

We further validate our theory by training classifiers from scratch for an unconditional diffusion model with 1000 timesteps and with 10 timesteps on the CIFAR-10 dataset (Krizhevsky, 2009). As shown in Figure 4, the upper 5 rows display results generated with classifiers corresponding to all 1000 diffusion timesteps (trained with batch size 64 for 100k iterations), while the lower 5 rows

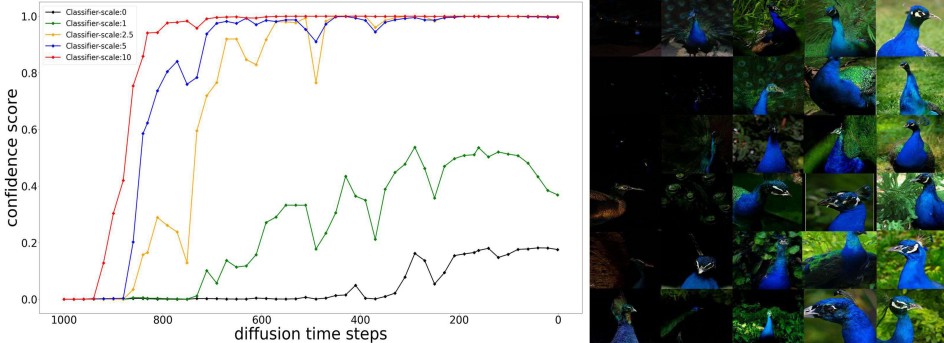

Figure 5: Left: Confidence score of classifiers with different classifiers scales from 0.0 to 10.0: Given the bias $\boldsymbol{\mu} = -0.03 \cdot \mathbb{1}$ on the initial sampling, the generated images for class "peacock".

visualize results guided by classifiers corresponding to only 10 timesteps (trained for 30k iterations). The quantitative results are reported in Table 2. Both models achieve comparable image quality, but our approach substantially reduces computational cost by requiring far fewer classifiers.

## 4.3 Verifying Contractive Property

Theorem 3.3 investigates the contractive property of the reverse dynamics (2) under the smoothness assumption of the unconditional score function and the strong log-concavity of the classifier. To test its validity, we first conduct a synthetic experiment on a toy dataset by setting the initial distribution as either a standard Gaussian or an arbitrary Gaussian, as shown in Appendix D.1.

To verify the contraction property discussed in Theorem 3.3 on real-world datasets, we test the conditional diffusion model by incrementally increasing the classifier guidance scale. The motivation is that, under the assumptions of Theorem B.1, i.e., the smoothness of the unconditional score function and the strong log-concavity of the classifier, increasing the guidance scale to a suitable value can make the contractive inequality hold, as discussed in Remark 3.

In this case, we initially samples $\boldsymbol{x}_T \sim \mathcal{N}(\boldsymbol{x}_T; \boldsymbol{\mu}, I)$ with the bias $\boldsymbol{\mu} = -0.03 \cdot \mathbb{1}$.

As shown in Figure 5, the classifier guidance scale increases from 0.0 to 10.0 from left to right. When the scale is 0, the samples generated by the conditional diffusion model without guidance are of poor quality. As the scale increases, the generated images gradually improve, ranging from nearly blank outputs to realistic bird images. This validates the contractive property of the reverse dynamics (2) even on real-world datasets. Moreover, this contractive property enhances the robustness of the diffusion model against distribution shifts in the initial sampling. Additional experimental results are provided in Appendix D.2.

## 4.4 Comparison to Classifier-free Guidance Model

Our idea can be directly applied to classifier-free guidance model (CFG), because CFG and classifier guidance model (CG) are theoretically equivalent. Note that the main goal of these two approaches is to estimate the guidance term $\nabla_{\boldsymbol{x}} \log p_t(y \mid \boldsymbol{x})$. In classifier guidance, a time-dependent classifier is trained to approximate $p_t(y \mid \cdot)$ on all noisy data. In classifier-free guidance, a new neural network $\boldsymbol{s}_\theta(t, \boldsymbol{x}, y)$ is trained to estimate the conditional score $\nabla_{\boldsymbol{x}} \log p_t(\boldsymbol{x} \mid y)$, while $\boldsymbol{s}_\phi(t, \boldsymbol{x}, \emptyset)$ approximates the unconditional score $\nabla_{\boldsymbol{x}} \log p_t(\boldsymbol{x})$. The guidance term is then computed as

$$\nabla_{\boldsymbol{x}} \log p_t(y \mid \boldsymbol{x}) \approx \boldsymbol{s}_\phi(t, \boldsymbol{x}, y) - \boldsymbol{s}_\phi(t, \boldsymbol{x}, \emptyset).$$

So the main goal of CFG is to train time-dependent $\boldsymbol{s}_\phi(t, \boldsymbol{x}, y)$. Based on our theoretical analysis, in stead of training a time-dependent $\boldsymbol{s}_\phi(\boldsymbol{x}_t, t, y)$, we can train $k$ time-independent

$$\boldsymbol{s}_{\phi_1}(\boldsymbol{x}_{t_1}, y), \cdots, \boldsymbol{s}_{\phi_k}(\boldsymbol{x}_{t_k}, y),$$

because according to Proposition 3.4, we only need guidance at $t_1, \cdots, t_k$, i.e., training $\boldsymbol{s}_{\phi_k}(\boldsymbol{x}, y)$ such that

$$\nabla_{\boldsymbol{x}} \log p_{t_i}(y \mid \boldsymbol{x}) \approx \boldsymbol{s}_{\phi_i}(\boldsymbol{x}, y) - \boldsymbol{s}_{\phi_i}(\boldsymbol{x}, \emptyset).$$

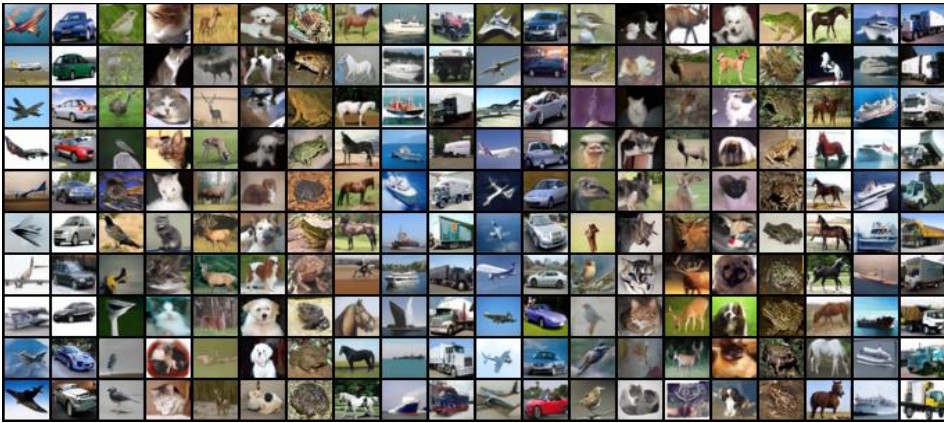

Figure 6: Class-conditional samples generated by our classifier-free guided diffusion model using only $k = 10$ conditional score $s_{\phi_i}(\boldsymbol{x}_{t_i}, y)$ with guidance scale $s = 1.3$ at selected timesteps $t_k \in [0, 100, 200, ...900]$.

For sampling, we only need to replace $\nabla_{\boldsymbol{x}} \log p_{t_i}(y \mid \boldsymbol{x})$ by $\boldsymbol{s}_{\phi_i}(\boldsymbol{x}, y) - \boldsymbol{s}_{\phi_i}(\boldsymbol{x}, \emptyset)$ in Algorithm 1.

Here, we demonstrate the qualitative results obtained by our classifier-free guidance diffusion model using $k = 10$ learned guidance heads. As shown in Figure 6, our approach is able to generate visually compelling class-conditional samples across a wide range of CIFAR-10 categories. Even though guidance is only provided at a limited set of time steps, the model still achieves strong semantic control while preserving high-frequency image details.

Compared with standard classifier-free guidance, our method significantly reduces the complexity of conditional score learning. Instead of modeling the conditional distribution over the entire diffusion trajectory, guidance networks are only instantiated at the selected key times $\{t_i\}_{i=1}^k$, thus lowering both training cost and memory usage. This supports our theoretical conclusion that the guidance signal does not need to be time-dense to effectively steer the reverse diffusion process.

In summary, these results validate that our proposed sparse-time guidance strategy is compatible with classifier-free guidance (CFG), and that only a few conditional score function are sufficient to achieve high-quality, class-consistent generation.

## 5 CONCLUSION

This paper explores the possibility of training a time-independent classifier to guide an unconditional diffusion model in generating target conditional distributions. We theoretically show that a single time-independent classifier trained on clean data can enable conditional generation under certain conditions. However, since real-world data often fails to satisfy these conditions, we propose two techniques to address this limitation. First, the initial sampling condition is intractable; we resolve this by simplifying the initialization to a standard Gaussian through analysis of the contractive property of the reverse process guided by a suitable classifier. Second, due to the manifold hypothesis, a single classifier lacks sufficient information for guidance; therefore, we employ a small number of time-independent classifiers trained at different noise levels to guide conditional generation on real-world data. To analyze the effect of the number of classifiers, we provide a theoretical convergence analysis and establish an upper bound in terms of the number of classifiers. Moreover, experiments on both synthetic and real-world datasets confirm our conclusion that using only a few time-independent classifiers achieves performance comparable to CGDM, which requires a time-dependent classifier trained on noisy data at all timesteps.

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

# A PROOFS

## A.1 PROOF OF THEOREM 3.1

*Proof.* If $\boldsymbol{x}_t \sim \tilde{p}_t(\boldsymbol{x}_t \mid y) \propto p_t(\boldsymbol{x}_t) h_y(\boldsymbol{x}_t)$ and $\tilde{p}(\boldsymbol{x}_{t-1} \mid \boldsymbol{x}_t, y)$ is defined as the theorem, then $\tilde{p}_{t-1}(\boldsymbol{x}_{t-1} \mid y)$ is

$$
\int \tilde{p}_t(\boldsymbol{x}_t \mid y)\tilde{p}(\boldsymbol{x}_{t-1} \mid \boldsymbol{x}_t, y)d\boldsymbol{x}_t \propto \int p_t(\boldsymbol{x}_t) h_y(\boldsymbol{x}_t) p(\boldsymbol{x}_{t-1} \mid \boldsymbol{x}_t)\frac{h_y(\boldsymbol{x}_{t-1})}{h_y(\boldsymbol{x}_t)}d\boldsymbol{x}_t
$$

$$
= \int p_t(\boldsymbol{x}_t) p(\boldsymbol{x}_{t-1} \mid \boldsymbol{x}_t) h_y(\boldsymbol{x}_{t-1})d\boldsymbol{x}_t
$$

$$
= h_y(\boldsymbol{x}_{t-1}) \int p_t(\boldsymbol{x}_t) p(\boldsymbol{x}_{t-1} \mid \boldsymbol{x}_t)d\boldsymbol{x}_t
$$

$$
= p_{t-1}(\boldsymbol{x}_{t-1}) h_y(\boldsymbol{x}_{t-1}).
$$

So $\boldsymbol{x}_{t-1} \sim \tilde{p}_{t-1}(\boldsymbol{x}_{t-1} \mid y) \propto p_{t-1}(\boldsymbol{x}_{t-1}) h_y(\boldsymbol{x}_{t-1})$. □

## A.2 PROOF OF PROPOSITION 3.2

*Proof.* First, by the reverse process of the unconditional diffusion model, we know

$$
p(\boldsymbol{x}_{t-1} \mid \boldsymbol{x}_t) = \mathcal{N}(\boldsymbol{x}_{t-1}; \boldsymbol{\mu}_t(\boldsymbol{x}_t), \sigma_t^2 I) \propto \exp(-\frac{\|\boldsymbol{x}_{t-1} - \boldsymbol{\mu}_t(\boldsymbol{x}_t)\|^2}{2\sigma_t^2}).
$$

Because the transition probability is

$$
\tilde{p}(\boldsymbol{x}_{t-1} \mid \boldsymbol{x}_t, y) = p(\boldsymbol{x}_{t-1} \mid \boldsymbol{x}_t)\frac{h_y(\boldsymbol{x}_{t-1})}{h_y(\boldsymbol{x}_t)} = p(\boldsymbol{x}_{t-1} \mid \boldsymbol{x}_t)\exp(\log h_y(\boldsymbol{x}_{t-1}) - \log h_y(\boldsymbol{x}_t))
$$

and by the Taylor formula at $\boldsymbol{x}_t$,

$$
\log h_y(\boldsymbol{x}_{t-1}) - \log h_y(\boldsymbol{x}_t) \approx (\boldsymbol{x}_{t-1} - \boldsymbol{x}_t)\nabla_{\boldsymbol{x}} \log h_y(\boldsymbol{x}_t)
$$

we can get

$$\tilde{p}(\boldsymbol{x}_{t-1} \mid \boldsymbol{x}_t, y) \propto \exp\left(-\frac{\|\boldsymbol{x}_{t-1} - \boldsymbol{\mu}_t(\boldsymbol{x}_t)\|^2}{2\sigma_t^2} + (\boldsymbol{x}_{t-1} - \boldsymbol{x}_t)\nabla_{\boldsymbol{x}} \log h_y(\boldsymbol{x}_t)\right)$$

$$\propto \exp(-\frac{\|\boldsymbol{x}_{t-1} - \boldsymbol{\mu}_t(\boldsymbol{x}_t) - \sigma_t^2 \nabla_{\boldsymbol{x}} \log h_y(\boldsymbol{x}_t)\|^2}{2\sigma_t^2}).$$

Therefore, we have $\boldsymbol{x}_{t-1} = \boldsymbol{\mu}(\boldsymbol{x}_t) + \sigma_t^2 \nabla_{\boldsymbol{x}} \log h_y(\boldsymbol{x}_t) + \sigma_t \boldsymbol{\varepsilon}, \ \boldsymbol{\varepsilon} \sim \mathcal{N}(0, I)$. $\square$

### A.3 PROOF OF PROPOSITION 3.4

*Proof.* Similarly as the proof in Theorem 3.1, if $\boldsymbol{x}_{t_i+1} \sim p_{t_i+1}(\boldsymbol{x}_{t_i+1})p_{t_i}(y \mid \boldsymbol{x}_{t_i+1})$, then with the transition probability $\tilde{p}_{t_i}(\boldsymbol{x}_{t_i} \mid \boldsymbol{x}_{t_i+1}, y)$ defined in (4), $\boldsymbol{x}_{t_i}$ obeys

$$\int p_{t_i+1}(\boldsymbol{x}_{t_i+1})p_{t_i}(y \mid \boldsymbol{x}_{t_i+1})p(\boldsymbol{x}_{t_i} \mid \boldsymbol{x}_{t_i+1})\frac{p_{t_i+1}(y \mid \boldsymbol{x}_{t_i})}{p_{t_i}(y \mid \boldsymbol{x}_{t_i+1})}d\boldsymbol{x}_{t_i+1}$$

$$= p_{t_{i+1}}(y \mid \boldsymbol{x}_{t_i}) \int p_{t_i+1}(\boldsymbol{x}_{t_i+1})p(\boldsymbol{x}_{t_i} \mid \boldsymbol{x}_{t_i+1})d\boldsymbol{x}_{t_i+1}$$

$$= p_{t_i}(\boldsymbol{x}_{t_i})p_{t_{i+1}}(y \mid \boldsymbol{x}_{t_i}).$$

So this transition probability is valid. And when viewing $p_t(y \mid \boldsymbol{x})$ as two variables function of $(t, \boldsymbol{x})$, by the Taylor formula,

$$\log p_{t_{i+1}}(y \mid \boldsymbol{x}_{t_i}) - \log p_{t_i}(y \mid \boldsymbol{x}_{t_i+1}) \approx (\boldsymbol{x}_{t_i} - \boldsymbol{x}_{t_i+1})\nabla_{\boldsymbol{x}} \log p_{t_i}(y \mid \boldsymbol{x}_{t_i+1}) + C,$$

where $C$ is independent with $\boldsymbol{x}_{t_i}$. Similarly as the proof of Proposition 3.2,

$$\tilde{p}_{t_i}(\boldsymbol{x}_{t_i} \mid \boldsymbol{x}_{t_i+1}, y) \propto \exp\left(-\frac{\|\boldsymbol{x}_{t_i} - \boldsymbol{\mu}(\boldsymbol{x}_{t_i+1})\|^2}{2\sigma_{t_i+1}^2} + (\boldsymbol{x}_{t_i} - \boldsymbol{x}_{t_i+1})\nabla_{\boldsymbol{x}} \log p_{t_i}(y \mid \boldsymbol{x}_{t_i+1})\right)$$

$$\propto \exp(-\frac{\|\boldsymbol{x}_{t_i} - \boldsymbol{\mu}(\boldsymbol{x}_{t_i+1}) - \sigma_{t_i+1}^2 \nabla_{\boldsymbol{x}} \log p_{t_i}(y \mid \boldsymbol{x}_{t_i+1})\|^2}{2\sigma_{t_i+1}^2}).$$

Therefore, generating

$$\boldsymbol{x}_{t_i} = \boldsymbol{\mu}(\boldsymbol{x}_{t_i+1}) + \sigma_{t_i+1}^2 \nabla_{\boldsymbol{x}} \log p_{t_i}(y \mid \boldsymbol{x}_{t_i+1}) + \sigma_{t_i+1}\boldsymbol{\varepsilon}. \qquad \square$$

## B PROOFS BY APPLYING SDE FORMULAS

Song et al. (2021) provided a stochastic differential equation (SDE) framework to explain the diffusion model, which is helpful for us to obtain the results of Theorem 3.3 and Theorem 3.5.

To simplify the following analysis, let's choose the forward process to be the Ornstein–Uhlenbeck (OU) process, for $t \in [0, T]$ (Note that here we use $T$ to be the endpoint of the diffusion interval, instead of the time-steps of discrete diffusion model, because in this section we won't consider the discrete version of the diffusion process),

$$d\boldsymbol{x}_t = -\boldsymbol{x}_t dt + \sqrt{2}dB_t, \quad \boldsymbol{x}_0 \sim p,$$

where $(B_t)_{t \in [0,T]}$ is a standard Brownian motion on $\mathbb{R}^d$. So it is a particular case of our above practical settings, by

$$\beta_t \equiv 2, \quad \forall\, t \in [0, T].$$

The OU process has an analytic solution

$$\boldsymbol{x}_t \stackrel{d}{=} \lambda_t \boldsymbol{x}_0 + \sigma_t W, \quad W \sim \mathcal{N}(0, I),$$

with $\lambda_t = e^{-t}$ and $\sigma_t = \sqrt{1 - e^{-2t}}$, where $\stackrel{d}{=}$ means the random variables of the RHS and the LHS have the same distribution function.

Now, to be more clear in notations, let denote $(\bar{\boldsymbol{x}}_t)_{t \in [0,T]}$ be the reverse process, that is,

$$\bar{\boldsymbol{x}}_t := \boldsymbol{x}_{T-t}.$$

Then $(\bar{\boldsymbol{x}}_t)_{t\in[0,T]}$ satisfies the SDE

$$d\bar{\boldsymbol{x}}_t = (\bar{\boldsymbol{x}}_t + 2\nabla_{\boldsymbol{x}} \log p_{T-t}(\bar{\boldsymbol{x}}_t))\, dt + \sqrt{2}d\bar{B}_t, \quad \bar{\boldsymbol{x}}_0 \sim p_T, \tag{5}$$

where $(\bar{B}_t)_{t\in[0,T]}$ is the Brownian motion in reverse time, and $p_t = \mathrm{Law}(\boldsymbol{x}_t)$, the density function of $\boldsymbol{x}_t$.

By Song et al. (2021), we can use the reverse process to generate the conditional distribution $p(\boldsymbol{x} \mid y)$ by replacing $\nabla_{\boldsymbol{x}} \log p_{T-t}(\boldsymbol{x})$ with

$$\nabla_{\boldsymbol{x}} \log p_{T-t}(\boldsymbol{x} \mid y) = \nabla_{\boldsymbol{x}} \log p_{T-t}(\boldsymbol{x}) + \nabla_{\boldsymbol{x}} \log p_{T-t}(y \mid \boldsymbol{x}).$$

Therefore, let $(\boldsymbol{y}_t)_{t\in[0,T]}$ be the conditional reverse process for generating $p(\boldsymbol{x} \mid y)$, so it satisfies the following SDE, for $t \in [0,T]$,

$$d\boldsymbol{y}_t = (\boldsymbol{y}_t + 2\nabla_{\boldsymbol{x}} \log p_{T-t}(\boldsymbol{y}_t) + 2\nabla_{\boldsymbol{x}} \log h_y(t, \boldsymbol{y}_t))\, dt + \sqrt{2}d\bar{B}_t, \quad \boldsymbol{y}_0 \sim p_T(\cdot \mid y), \tag{6}$$

where $h_y(t, \boldsymbol{x}) := p_{T-t}(y \mid \boldsymbol{x})$. Along this process, it can generate $\boldsymbol{y}_T \sim p(\cdot \mid y) =: p_y$.

## B.1 CONTRACTIVE PROPERTY

In our setting, instead of choosing $h_y(t, \boldsymbol{x}) = p_{T-t}(y \mid \boldsymbol{x})$, we let

$$h_y(\boldsymbol{x}) = p_0(y \mid \boldsymbol{x}),$$

which is a time-independent classifier trained the clean dataset. Therefore, the SDE formula of our reverse process (2) is defined as

$$d\bar{\boldsymbol{y}}_t = (\bar{\boldsymbol{y}}_t + 2\nabla_{\boldsymbol{x}} \log p_{T-t}(\bar{\boldsymbol{y}}_t) + 2\nabla_{\boldsymbol{x}} \log h_y(\bar{\boldsymbol{y}}_t))\, dt + \sqrt{2}dB_t, \quad \bar{\boldsymbol{y}}_0 \sim Z_T p_T(\cdot)p_0(y \mid \cdot).$$

*Remark* 1. In practice, there exists a scale $s$ to control the strength of guidance, that is replacing $\nabla_{\boldsymbol{x}} \log h_y(\bar{\boldsymbol{y}}_t)$ by $s\nabla_{\boldsymbol{x}} \log h_y(\bar{\boldsymbol{y}}_t)$. Here to simplify the analysis, we set $s = 1$.

As mentioned before, it is intractable to draw $\bar{\boldsymbol{y}}_0 \sim Z_T p_T(\cdot)p_0(y \mid \cdot)$ or its approximated version $\bar{\boldsymbol{y}}_0 \sim Z_T \mathcal{N}(0, I)p_0(y \mid \cdot)$. Instead, we consider

$$d\hat{\boldsymbol{y}}_t = (\hat{\boldsymbol{y}}_t + 2\nabla_{\boldsymbol{x}} \log p_{T-t}(\hat{\boldsymbol{y}}_t) + 2\nabla_{\boldsymbol{x}} \log h_y(\hat{\boldsymbol{y}}_t))\, dt + \sqrt{2}dB_t, \quad \hat{\boldsymbol{y}}_0 \sim \mathcal{N}(0, I).$$

The problem is what is the distance between $\bar{\boldsymbol{y}}_T \sim \bar{p}_y$ and $\hat{\boldsymbol{y}}_T \sim \hat{p}_y$. Here we choose the Wasserstein distance to measure the distance, which coincides with the FID score in practice. For two distribution $p, q$, the 2-Wasserstein distance between $p$ and $q$ is

$$\mathcal{W}_2(p, q)^2 = \inf\left\{\int \|x - y\|^2 d\gamma(x, y) \colon \gamma \in \Gamma(p, q)\right\}$$
$$= \inf\left\{\mathbb{E}\left[\|X - Y\|^2\right] \colon X \sim p,\ Y \sim q\right\},$$

where $\Gamma(p, q)$ is the set of all joint distributions with marginal distributions $p$ and $q$; see more details in Chewi et al. (2024).

**Assumption 1.** There exists $L_p > 0$ such that $\log p_t(\boldsymbol{x})$ is $L_p$-smooth, i.e. $\|\nabla_{\boldsymbol{x}}^2 \log p_t(\boldsymbol{x})\|_{\mathsf{op}} \leq L_p$.

*Remark* 2. The smoothness condition of score functions is widely used in theoretical analysis (Li & Yan, 2024; Chen et al., 2023b;c). In fact, it can be replaced by the smoothness condition of $\log p_0(\boldsymbol{x})$; see more details in Chen et al. (2023a).

Under Assumption 1, we can provide a formal version of Theorem 3.3 and its proof.

**Theorem B.1** (Formal). *Using notations as above and under Assumption 1, if for any $\boldsymbol{x}$*

$$-\nabla_{\boldsymbol{x}}^2 \log h_y(\boldsymbol{x}) \succeq MI,$$

*where $M > 0$ such that $M > L_p + 1/2$, i.e. $h_y$ is $M$-strongly log-concave, then*

$$\mathcal{W}_2(\bar{p}_y, \hat{p}_y) \leq \mathcal{O}\left(e^{-T}\right).$$

*Proof.* First, let $K = 4(M - L_p) - 2 > 0$. By Itô's formula,

$$d\left(\|\bar{\boldsymbol{y}}_t - \hat{\boldsymbol{y}}_t\|^2 e^{Kt}\right) = Ke^{Kt}\|\bar{\boldsymbol{y}}_t - \hat{\boldsymbol{y}}_t\|^2 dt + 2e^{Kt}\langle\bar{\boldsymbol{y}}_t - \hat{\boldsymbol{y}}_t, d\bar{\boldsymbol{y}}_t - d\hat{\boldsymbol{y}}_t\rangle$$

$$\begin{aligned}
&= (K+2)e^{Kt}\|\bar{\boldsymbol{y}}_t - \hat{\boldsymbol{y}}_t\|^2 dt \\
&\quad + 4e^{Kt}\langle\bar{\boldsymbol{y}}_t - \hat{\boldsymbol{y}}_t, \nabla_{\boldsymbol{x}}\log p_{T-t}(\bar{\boldsymbol{y}}_t) - \nabla_{\boldsymbol{x}}\log p_{T-t}(\hat{\boldsymbol{y}}_t)\rangle\, dt \\
&\quad + 4e^{Kt}\langle\bar{\boldsymbol{y}}_t - \hat{\boldsymbol{y}}_t, \nabla_{\boldsymbol{x}}\log h_y(\bar{\boldsymbol{y}}_t) - \nabla_{\boldsymbol{x}}\log h_y(\hat{\boldsymbol{y}}_t)\rangle\, dt.
\end{aligned} \tag{7}$$

Under Assumption 1, because $\|\nabla_{\boldsymbol{x}}^2 \log p_t(\boldsymbol{x})\|_{\mathsf{op}} \leq L_p$, we have

$$\langle\bar{\boldsymbol{y}}_t - \hat{\boldsymbol{y}}_t, \nabla_{\boldsymbol{x}}\log p_{T-t}(\bar{\boldsymbol{y}}_t) - \nabla_{\boldsymbol{x}}\log p_{T-t}(\hat{\boldsymbol{y}}_t)\rangle \leq L_p\|\bar{\boldsymbol{y}}_t - \hat{\boldsymbol{y}}_t\|^2. \tag{8}$$

Moreover, because $\nabla_{\boldsymbol{x}}^2 \log h_y(\boldsymbol{x}) \preceq -MI$,

$$\langle\bar{\boldsymbol{y}}_t - \hat{\boldsymbol{y}}_t, \nabla_{\boldsymbol{x}}\log h_y(\bar{\boldsymbol{y}}_t) - \nabla_{\boldsymbol{x}}\log h_y(\hat{\boldsymbol{y}}_t)\rangle \leq -M\|\bar{\boldsymbol{y}}_t - \hat{\boldsymbol{y}}_t\|^2. \tag{9}$$

By Combining (8) and (9) with (7), we obtain

$$d\left(\|\bar{\boldsymbol{y}}_t - \hat{\boldsymbol{y}}_t\|^2 e^{Kt}\right) \leq (K + 2 + 4(L_p - M))dt = 0,$$

which implies that

$$\|\bar{\boldsymbol{y}}_t - \hat{\boldsymbol{y}}_t\|^2 \leq e^{-Kt}\|\bar{\boldsymbol{y}}_0 - \hat{\boldsymbol{y}}_0\|^2 \Rightarrow \mathbb{E}\left[\|\bar{\boldsymbol{y}}_t - \hat{\boldsymbol{y}}_t\|^2\right] \leq e^{-Kt}\mathbb{E}\left[\|\bar{\boldsymbol{y}}_0 - \hat{\boldsymbol{y}}_0\|^2\right].$$

Therefore,

$$\mathcal{W}_2(\bar{p}_y, \hat{p}_y)^2 \leq \mathbb{E}\left[\|\bar{\boldsymbol{y}}_T - \hat{\boldsymbol{y}}_T\|^2\right] \leq Ce^{-KT} = \mathcal{O}\left(e^{-T}\right). \qquad \square$$

*Remark* 3. Note that even $h_y(\boldsymbol{x})$ is $M$-strongly log-concave, we cannot guarantee $M > L_p + 1/2$. However, as mentioned before, in practice, we can adjust the guidance scale $s$ to make $-s\nabla_{\boldsymbol{x}}^2 \log h_y(\boldsymbol{x}) \succeq sMI$ and $sM > L_p + 1/2$ such that the contractive property can be satisfied; see the experiments in Section 4.3 and Appendix D.2.

## B.2 CHOICE OF THE NUMBER OF CLASSIFIERS

In the following, we provide a theoretical analysis of the relationship between the performance and the number of classifiers $k$.

In our settings, it first chooses a partition of $[0, T]$,

$$0 = t_0 < t_1 < \cdots < t_k = T,$$

and $t_{i+1} - t_i = T/k$ for any $i = 0, 1, \cdots, k-1$. Then define $\tilde{h}_y(t, \boldsymbol{x})$ piecewise as

$$\tilde{h}_y(t, \boldsymbol{x}) = h_y(t_i, \boldsymbol{x}), \quad \forall\, t \in (t_i, t_{i+1}].$$

So the generated process $(\tilde{\boldsymbol{y}}_t)_{t \in [0,T]}$ in our case satisfies the following SDE

$$d\tilde{\boldsymbol{y}}_t = \left(\tilde{\boldsymbol{y}}_t + 2\nabla_{\boldsymbol{x}}\log p_{T-t}(\tilde{\boldsymbol{y}}_t) + 2\nabla_{\boldsymbol{x}}\log\tilde{h}_y(t, \tilde{\boldsymbol{y}}_t)\right)dt + \sqrt{2}d\bar{B}_t, \quad \tilde{\boldsymbol{y}}_0 \sim p_T(\cdot \mid y). \tag{10}$$

Let $\tilde{\boldsymbol{y}}_T \sim \tilde{p}_y$. So the main goal is to measure the total variation of $p_y$ and generated $\tilde{p}_y$, $\mathsf{TV}(p_y, \tilde{p}_y)$, from the SDEs (6) and (10). Motivated by Bortoli et al. (2021); Chen et al. (2023c), we will apply Girsanov's theorem to this problem. Therefore, in the following two subsections, we will first introduce Girsanov's theorem and explain how it can be applied to this kind of problem. Then we will use the results to analyze the upper bound of $\mathsf{TV}(p_y, \tilde{p}_y)$ with respect to the number of classifiers, $k$. First, we need the following three assumptions.

**Assumption 2.** We assume that $\mathsf{m}_2^2 := \mathbb{E}^{p(\cdot|y)}\left[\|\cdot\|^2\right] = \mathbb{E}\left[\|\boldsymbol{y}_T\|^2\right] < \infty$.

**Assumption 3.** For all $t \in [0, T]$, $\log p_t(\boldsymbol{x} \mid y)$ is $L$-smooth for some $L \geq 1$, that is, $\|\nabla^2 \log p_t(\boldsymbol{x} \mid y)\|_{\mathsf{op}} \leq L$.

**Assumption 4.** There is an $A > 0$ such that for all $t \in [0, T]$, $\|\partial_t \nabla_{\boldsymbol{x}}\log h_y(t, \boldsymbol{x})\| \leq A\|\boldsymbol{x}\|$.

Note that $\nabla^2 u$ means the Hessian of $u$ and $\|\cdot\|_{\mathsf{op}}$ is the operator norm of a matrix.

*Remark* 4. (1) Assumption 2 is appeared in many works, such as Li & Yan (2024); Chen et al. (2023b;c). But it can be replaced by the bounded support of $p$ (Huang et al., 2024), or $L$-smoothness of $\log p_0(\cdot \mid y)$ (Chen et al., 2023a).

(2) Assumption 3 is not weird because when analyzing SDE (5), it usually assumes the $L$-smoothness of $\log p_t$, such as Chen et al. (2023b;c). Here we just replace $\log p_t$ with $\log p_t(\cdot \mid y)$ for analyzing SDE (6).

**Theorem B.2.** *Using the notations as above and under the Assumption 2, 3, 4, there is a constant* $C = C(L, d, T, \mathfrak{m}_2) > 0$ *such that*

$$\mathsf{TV}(p_y, \tilde{p}_y) \leq \frac{C}{k}.$$

### B.2.1 GIRSANOV'S THEOREM AND APPROXIMATED TECHNIQUE

In this section, let's fix a probability space $(\Omega, \mathcal{F}, \mathbb{P})$ and a Brownian motion $B = (B_t)_{t \in [0,T]}$, or called a $\mathbb{P}$-Brownian motion. Besides, let the probability space be equipped with the natural filtration induced by $B$.

**Theorem B.3** (Girsanov's Theorem, Theorem 6.3 of Liptser & Shiryaev (2013)). *For $t \in [0, T]$, let* $M_t = \int_0^t \theta_u dB_u$ *where $B$ is a $\mathbb{P}$-Brownian motion. Assume that $\theta_t \in \mathcal{L}^2(B)$, that is*

$$\mathbb{E}^{\mathbb{P}} \left[ \frac{1}{2} \int_0^T \|\theta_t\|^2 dt \right] < \infty.$$

*Then $M$ is a $\mathbb{P}$-martingale in $\mathcal{L}^2(\mathbb{P})$. Moreover, if $\mathbb{E}^{\mathbb{P}}[\mathcal{E}(M)_T] = 1$, where*

$$\mathcal{E}(M)_t := \exp \left( \int_0^t \theta_u dB_u - \frac{1}{2} \int_0^t \|\theta_u\|^2 du \right).$$

*Then the process*

$$t \mapsto B_t - \int_0^t \theta_u du,$$

*is a $\mathbb{Q}$-Brownian motion for*

$$\frac{d\mathbb{Q}}{d\mathbb{P}} = \mathcal{E}(M)_T = \exp \left( \int_0^T \theta_t dB_t - \frac{1}{2} \int_0^T \|\theta_t\|^2 dt \right).$$

Girsanov's theorem can be applied to analyze the behaviors of two SDEs with different drifts and the same noise scale. The following lemma explicitly shows that. This result appeared in Bortoli et al. (2021); Chen et al. (2023c), but they proved it in the path-space (Wiener space). Here we provide another proof without considering the Wiener space and the Wiener measure.

**Lemma B.4.** *Considering the following two SDEs,*

$$dX_t^{(1)} = b_t^{(1)}(X_t^{(1)})dt + \sqrt{2}dB_t, \quad X_0^{(1)} \sim \rho_0,$$
$$dX_t^{(2)} = b_t^{(2)}(X_t^{(2)})dt + \sqrt{2}dB_t, \quad X_0^{(2)} \sim \rho_0,$$

*and let*

$$\theta_t = \frac{1}{\sqrt{2}} \left( b_t^{(1)} - b_t^{(2)} \right).$$

*Assume the conditions in the above theorem are satisfied,* i.e.

$$\mathbb{E}^{\mathbb{P}} \left[ \frac{1}{2} \int_0^T \|\theta_t\|^2 dt \right] < \infty, \quad \mathbb{E}^{\mathbb{P}} \left[ \exp \left( \int_0^T \theta_t dB_t - \frac{1}{2} \int_0^T \|\theta_t\|^2 dt \right) \right] = 1.$$

*Let $\mu^{(i)} = (X_T^{(i)})_{\#}\mathbb{P}$ be the distribution of $X_T^{(i)}$ for $i = 1, 2$. Then we have*

$$\mathsf{TV}^2(\mu^{(1)}, \mu^{(2)}) \leq \mathsf{KL}(\mu^{(1)} \| \mu^{(2)}) \leq \frac{1}{4} \int_0^T \mathbb{E}^{\mathbb{P}} \left[ \|b_t^{(1)} - b_t^{(2)}\|^2 \right] dt.$$

*Proof.* For

$$\theta_t = \frac{1}{\sqrt{2}} \left( b_t^{(1)}(X_t^{(2)}) - b_t^{(2)}(X_t^{(2)}) \right),$$

because it satisfies the conditions in Girsanov's theorem, there is a new probability measure $\mathbb{Q}$ such that

$$W_t = B_t - \frac{1}{\sqrt{2}} \int_0^t \left( b_u^{(1)}(X_u^{(2)}) - b_u^{(2)}(X_u^{(2)}) \right) du.$$

is a $\mathbb{Q}$-Brownian motion. So

$$\sqrt{2}dB_t = \sqrt{2}dW_t + \left( b_t^{(1)}(X_t^{(2)}) - b_t^{(2)}(X_t^{(2)}) \right) dt.$$

Then replacing $dB_t$ by $dW_t$ in $X^{(2)}$'s equation, we have

$$dX_t^{(2)} = b_t^{(2)}(X_t^{(2)})dt + \sqrt{2}dB_t = b_t^{(1)}(X_t^{(2)})dt + \sqrt{2}dW_t.$$

By comparing this equation with the equation of $X^{(1)}$ *w.s.t.* $dB_t$,

$$dX_t^{(1)} = b_t^{(1)}(X_t^{(1)})dt + \sqrt{2}dB_t, \quad X_0^{(1)} \sim \rho_0,$$
$$dX_t^{(2)} = b_t^{(1)}(X_t^{(2)})dt + \sqrt{2}dW_t, \quad X_0^{(2)} \sim \rho_0,$$

we can see they have the same formula when considering $X^{(1)}$ on $(\Omega, \mathbb{P})$ and $X^{(2)}$ on $(\Omega, \mathbb{Q})$. Therefore, by the uniqueness of the solution of SDE (Liptser & Shiryaev, 2013),

$$\mu^{(1)} = (X_T^{(1)})_{\#}\mathbb{P},$$
$$\mu^{(2)} = (X_T^{(2)})_{\#}\mathbb{P} = (X_T^{(1)})_{\#}\mathbb{Q},$$

and we have

$$\mathrm{KL}(\mu^{(1)}\|\mu^{(2)}) = \mathrm{KL}\left( (X_T^{(1)})_{\#}\mathbb{P} \,\middle\|\, (X_T^{(1)})_{\#}\mathbb{Q} \right).$$

By the following Lemma B.5, this implies that

$$\mathrm{KL}(\mu^{(1)}\|\mu^{(2)}) \le \mathrm{KL}(\mathbb{P}\|\mathbb{Q}).$$

To calculate the right-hand side, by Girsanov's theorem, we have

$$\frac{d\mathbb{Q}}{d\mathbb{P}} = \exp\left( \int_0^T \theta_t dB_t - \frac{1}{2} \int_0^T \|\theta_t\|^2 dt \right) \Rightarrow \frac{d\mathbb{P}}{d\mathbb{Q}} = \exp\left( -\int_0^T \theta_t dB_t + \frac{1}{2} \int_0^T \|\theta_t\|^2 dt \right).$$

Therefore,

$$\mathrm{KL}(\mu^{(1)}\|\mu^{(2)}) \le \mathrm{KL}(\mathbb{P}\|\mathbb{Q}) = \mathbb{E}^{\mathbb{P}}\left[ \log \frac{d\mathbb{P}}{d\mathbb{Q}} \right]$$

$$= \mathbb{E}^{\mathbb{P}}\left[ -\int_0^T \theta_t dB_t + \frac{1}{2} \int_0^T \|\theta_t\|^2 dt \right]$$

$$= \frac{1}{4} \int_0^T \mathbb{E}^{\mathbb{P}}\left[ \|b_t^{(1)}(X_t^{(2)}) - b_t^{(2)}(X_t^{(2)})\|^2 \right] dt.$$

by the fact that $\mathbb{E}^{\mathbb{P}}\left[ -\int_0^T \theta_t dB_t \right] = 0$. Finally, by Pinsker's inequality, $\mathsf{TV}^2 \le \mathrm{KL}$. □

The following lemma is basically a particular case of data processing inequality (Lemma 9.4.5 in Ambrosio et al. (2008)). Here we provide easy proof for the sake of completeness.

**Lemma B.5.** *Let $(\Omega, \mathcal{F})$ be a measurable space and $\mathbb{P} \ll \mathbb{Q}$ be two probability measures on it. Let $X : \Omega \to \mathbb{R}^d$ be a random variable with $\mathbb{P}_X = X_{\#}\mathbb{P}, \mathbb{Q}_X = X_{\#}\mathbb{Q}$. Then we have*

$$\mathrm{KL}(\mathbb{P}_X\|\mathbb{Q}_X) \le \mathrm{KL}(\mathbb{P}\|\mathbb{Q}).$$

*Proof.* First, $\mathbb{P} \ll \mathbb{Q}$ implies $\mathbb{P}_X \ll \mathbb{Q}_X$ by definition. Next, we prove that

$$\mathbb{E}^{\mathbb{Q}}\left[\frac{d\mathbb{P}}{d\mathbb{Q}} \,\middle|\, \sigma(X)\right] = \frac{d\mathbb{P}_X}{d\mathbb{Q}_X} \circ X.$$

First, because $\mathbb{E}^{\mathbb{Q}}\left[\frac{d\mathbb{P}}{d\mathbb{Q}} \,\middle|\, \sigma(X)\right]$ is $\sigma(X)$-measurable, there is a measurable function $h\colon \mathbb{R}^d \to \mathbb{R}$ such that $\mathbb{E}^{\mathbb{Q}}\left[\frac{d\mathbb{P}}{d\mathbb{Q}} \,\middle|\, \sigma(X)\right] = h(X)$. Then for any $B \in \mathcal{R}^d$ (Borel sets of $\mathbb{R}^d$),

$$\int_{\mathbb{R}^d} \mathbb{1}_B \frac{d\mathbb{P}_X}{d\mathbb{Q}_X} d\mathbb{Q}_X = \int_{\mathbb{R}^d} \mathbb{1}_B d\mathbb{P}_X$$

$$= \int_{\Omega} \mathbb{1}_B \circ X d\mathbb{P}$$

$$= \int_{\Omega} (\mathbb{1}_B \circ X) \frac{d\mathbb{P}}{d\mathbb{Q}} d\mathbb{Q}.$$

Clearly, $\mathbb{1}_B \circ X$ is $\sigma(X)$-measurable so

$$\mathbb{E}^{\mathbb{Q}}\left[(\mathbb{1}_B \circ X)\frac{d\mathbb{P}}{d\mathbb{Q}} \,\middle|\, \sigma(X)\right] = (\mathbb{1}_B \circ X)\mathbb{E}^{\mathbb{Q}}\left[\frac{d\mathbb{P}}{d\mathbb{Q}} \,\middle|\, \sigma(X)\right],$$

and by $\Omega \in \sigma(X)$,

$$\int_{\mathbb{R}^d} \mathbb{1}_B \frac{d\mathbb{P}_X}{d\mathbb{Q}_X} d\mathbb{Q}_X = \int_{\Omega} (\mathbb{1}_B \circ X)\frac{d\mathbb{P}}{d\mathbb{Q}} d\mathbb{Q}$$

$$= \int_{\Omega} \mathbb{E}^{\mathbb{Q}}\left[(\mathbb{1}_B \circ X)\frac{d\mathbb{P}}{d\mathbb{Q}} \,\middle|\, \sigma(X)\right] d\mathbb{Q}$$

$$= \int_{\Omega} (\mathbb{1}_B \circ X)\mathbb{E}^{\mathbb{Q}}\left[\frac{d\mathbb{P}}{d\mathbb{Q}} \,\middle|\, \sigma(X)\right] d\mathbb{Q}$$

$$= \int_{\Omega} (\mathbb{1}_B \circ X)(h \circ X) d\mathbb{Q}$$

$$= \int_{\mathbb{R}^d} \mathbb{1}_B h \, d\mathbb{Q}_X.$$

Therefore, $\mathbb{Q}_X$-surely we have $h = \frac{d\mathbb{P}_X}{d\mathbb{Q}_X}$ and thus we have the desired result. By this and the Jensen's inequality of the conditional expectation for the convex function $\eta(x) = (x \log x)\mathbb{1}_{(0,\infty)}(x)$ defined on $\mathbb{R}$, we have

$$\mathrm{KL}(\mathbb{P}_X \| \mathbb{Q}_X) = \int_{\mathbb{R}^d} \eta\left(\frac{d\mathbb{P}_X}{d\mathbb{Q}_X}\right) d\mathbb{Q}_X = \int_{\Omega} \eta\left(\frac{d\mathbb{P}_X}{d\mathbb{Q}_X} \circ X\right) d\mathbb{Q} = \int_{\Omega} \eta\left(\mathbb{E}^{\mathbb{Q}}\left[\frac{d\mathbb{P}}{d\mathbb{Q}} \,\middle|\, \sigma(X)\right]\right) d\mathbb{Q}$$

$$\leq \int_{\Omega} \mathbb{E}^{\mathbb{Q}}\left[\eta\left(\frac{d\mathbb{P}}{d\mathbb{Q}}\right) \,\middle|\, \sigma(X)\right] d\mathbb{Q} = \int_{\Omega} \eta\left(\frac{d\mathbb{P}}{d\mathbb{Q}}\right) d\mathbb{Q} = \mathrm{KL}(\mathbb{P}\|\mathbb{Q}).$$

$\square$

Lemma B.4 provides us with a method to measure the distance of generated distributions from two SDEs with different drifts. But in order to apply Girsanov's theorem, we need the following two conditions

$$\mathbb{E}^{\mathbb{P}}\left[\int_0^T \|\theta_t\|^2 dt\right] < \infty, \quad \mathbb{E}^{\mathbb{P}}[\mathcal{E}(M)_T] = 1.$$

For practical problems, the first condition is usually satisfied. But we cannot guarantee the second condition $\mathbb{E}^{\mathbb{P}}[\mathcal{E}(M)_T] = 1$, or equivalently $\mathcal{E}(M)$ a $\mathbb{P}$-martingale. So we use the approximation technique introduced in Chen et al. (2023c). Also, they considered this problem in the Wiener space. Here we slightly modify their proofs to omit their settings in the Wiener space.

**Lemma B.6.** *Let the settings be the same as Lemma B.4 but with only one assumption*

$$\mathbb{E}^{\mathbb{P}}\left[\int_0^T \|\theta_t\|^2 dt\right] = \frac{1}{2}\mathbb{E}^{\mathbb{P}}\left[\int_0^T \|b_t^{(1)} - b_t^{(2)}\|^2 dt\right] \leq M < \infty.$$

*Then it still has*

$$\mathsf{TV}^2(\mu^{(1)}, \mu^{(2)}) \leq M.$$

*Proof.* First, by the (3.4)Proposition of Chapter IV in Revuz & Yor (2013), we have known $\mathcal{E}(M)$ is a local martingale, which means there is a nondecreasing sequence of stopping times $T_n$ with the property $T_n \uparrow T$ such that $(\mathcal{E}(M)_{t \wedge T_n})_{t \in [0, T_n]}$ is a $\mathbb{P}$-martingale(see the (1.5)Definition of Chapter IV in Revuz & Yor (2013)). Besides, let $M^n = M^{T_n}$, that is

$$(M^n)_t := M_{t \wedge T_n} = \int_0^{t \wedge T_n} \theta_u du = \begin{cases} \int_0^t \theta_u dB_u, & t \leq T_n \\ \int_0^{T_n} \theta_u dB_u, & t > T_n \end{cases}$$

Therefore. by the definition of the exponential of a martingale,

$$\mathcal{E}(M^n)_t = \begin{cases} \exp\left(\int_0^t \theta_u dB_u - \frac{1}{2}\int_0^t \|\theta_u\|^2 du\right), & t \leq T_n \\ \exp\left(\int_0^{T_n} \theta_u dB_u - \frac{1}{2}\int_0^{T_n} \|\theta_u\|^2 du\right), & t > T_n \end{cases}$$

and so $\mathcal{E}(M^n)_t = \mathcal{E}(M)_{t \wedge T_n}$. Note that

$$M_t^n = \int_0^{t \wedge T_n} \theta_u dB_u = \int_0^t \theta_u \mathbb{1}_{t \in [0, T_n]} dB_u.$$

So martingale $M^n$ satisfies the conditions of Girsanov's Theorem. There is a probability measure $\mathbb{Q}^n$ on $\Omega$ such that

$$W_t^n = B_t - \int_0^t \theta_u \mathbb{1}_{t \in [0, T_n]} du = B_t - \frac{1}{\sqrt{2}} \int_0^t \left(b_u^{(1)} - b_u^{(2)}\right) \mathbb{1}_{t \in [0, T_n]}(u) du.$$

is a Brownian motion and we have

$$\frac{d\mathbb{Q}^n}{d\mathbb{P}} = \exp\left(\int_0^T \theta_t \mathbb{1}_{t \in [0, T_n]} dB_t - \frac{1}{2}\int_0^T \|\theta_t\|^2 \mathbb{1}_{t \in [0, T_n]} dt\right)$$

$$= \exp\left(\int_0^{T_n} \theta_t dB_t - \frac{1}{2}\int_0^{T_n} \|\theta_t\|^2 dt\right).$$

which implies

$$\mathrm{KL}\left(\mathbb{P}\|\mathbb{Q}^n\right) = \mathbb{E}^{\mathbb{P}}\left[\log \frac{d\mathbb{P}}{d\mathbb{Q}^n}\right] = \int_0^{T_n} \mathbb{E}^{\mathbb{P}}\left[\|\theta_t\|^2\right] dt$$

$$= \frac{1}{4}\int_0^{T_n} \mathbb{E}^{\mathbb{P}}\left[\|b_t^{(1)} - b_t^{(2)}\|^2\right] dt$$

$$\leq \frac{1}{4}\int_0^T \mathbb{E}^{\mathbb{P}}\left[\|b_t^{(1)} - b_t^{(2)}\|^2\right] dt \leq M.$$

Next, reconsidering the second SDE

$$dX_t^{(2)} = b_t^{(2)}(X_t^{(2)})dt + \sqrt{2}dB_t$$
$$= b_t^{(1)}(X_t^{(2)})\mathbb{1}_{t \in [0, T_n]}(t)dt + b_t^{(2)}(X_t^{(2)})\mathbb{1}_{t \in [T_n, T]}(t)dt + \sqrt{2}dW_t^n, \quad X_0^{(2)} \sim \rho_0.$$

and the equation

$$dX_t^n = b_t^{(1)}(X_t^n)\mathbb{1}_{t \in [0, T_n]}(t)dt + b_t^{(2)}(X_t^n)\mathbb{1}_{t \in [T_n, T]}(t)dt + \sqrt{2}dB_t, \quad X_0^n \sim \rho_0.$$

Let $\mu_n^{(1)}$ be the distribution of $X_T^n$ under $\mathbb{P}$. But we can see it has the same formula as $X_t^{(2)}$ under $\mathbb{Q}^n$. So

$$\mu^{(2)} = (X_T^{(2)})_{\#}\mathbb{P}$$
$$\mu_n^{(1)} = (X_T^n)_{\#}\mathbb{P} = (X_T^{(2)})_{\#}\mathbb{Q}^n.$$

And by the decreasing property of the relative entropy under the push-forward map,

$$\text{KL}\left(\mu^{(2)}\|\mu_n^{(1)}\right) \le \text{KL}\left(\mathbb{P}\|\mathbb{Q}^n\right) \le M.$$

Note that for all $t \le T_n$, $X_t^n = X_t^{(1)}$ by the uniqueness of solution of SDE. By the Lemma 13 in Chen et al. (2023c), for any $\varepsilon > 0$,

$$\left(X_{t\wedge(T-\varepsilon)}^n\right)_{t\in[0,T]} \to \left(X_{t\wedge(T-\varepsilon)}^{(1)}\right)_{t\in[0,T]} \quad a.s., \quad \text{as } n \to \infty.$$

Therefore, $X_{T-\varepsilon}^n \to X_{T-\varepsilon}^{(1)}$ a.s. as $n \to \infty$. Let $\mu_{n,\varepsilon}^{(1)} = (X_{T-\varepsilon}^n)_{\#}\mathbb{P}$ and $\mu_\varepsilon^{(1)} = (X_{T-\varepsilon}^{(1)})_{\#}\mathbb{P}$. Then for any continuous and bounded $f$ define on $\mathbb{R}^d$,

$$\int_{\mathbb{R}^d} f d\mu_{n,\varepsilon}^{(1)} = \int_\Omega f \circ X_{T-\varepsilon}^n d\mathbb{P} \to \int_\Omega f \circ X_{T-\varepsilon}^{(1)} d\mathbb{P} = \int_{\mathbb{R}^d} f d\mu_\varepsilon^{(1)}.$$

as $n \to \infty$, which means $\mu_{n,\varepsilon}^{(1)} \to \mu_\varepsilon^{(1)}$ weakly as $n \to \infty$. Besides, let $\mu_\varepsilon^{(2)} = (X_{T-\varepsilon}^{(2)})_{\#}\mathbb{P}$. Then by the lower semicontinuity of KL divergence (Lemma 9.4.3 in Ambrosio et al. (2008)),

$$\text{KL}\left(\mu_\varepsilon^{(2)}\|\mu_\varepsilon^{(1)}\right) \le \liminf_{n\to\infty} \text{KL}\left(\mu_\varepsilon^{(2)}\|\mu_{n,\varepsilon}^{(1)}\right).$$

Similarly as above, by comparing the equation in $W_t^n$ and $B_t$,

$$\mu_{n,\varepsilon}^{(1)} = (X_{T-\varepsilon}^n)_{\#}\mathbb{P} = (X_{T-\varepsilon}^{(2)})_{\#}\mathbb{Q}^n.$$

So we have

$$\text{KL}\left(\mu_\varepsilon^{(2)}\|\mu_\varepsilon^{(1)}\right) \le \liminf_{n\to\infty} \text{KL}\left((X_{T-\varepsilon}^{(2)})_{\#}\mathbb{P}\|(X_{T-\varepsilon}^{(2)})_{\#}\mathbb{Q}^n\right)$$

$$\le \liminf_{n\to\infty} \text{KL}\left(\mathbb{P}\|\mathbb{Q}^n\right)$$

$$\le M.$$

And because $X_{T-\varepsilon}^{(i)} \to X_T^{(i)}$ a.s. as $\varepsilon \to 0^+$, $\mu_\varepsilon^{(i)} \to \mu^{(i)}$ weakly for $i = 1, 2$. Using the same property, we have

$$\text{KL}(\mu^{(2)}\|\mu^{(1)}) \le \liminf_{\varepsilon\to 0^+} \text{KL}\left(\mu_\varepsilon^{(2)}\|\mu_\varepsilon^{(1)}\right) \le M.$$

Finally, by Pinsker's inequality, $\text{TV}^2 \le \text{KL}$. $\qquad\square$

### B.2.2 UPPER BOUND OF TOTAL VARIATION

*Proof of Theorem B.2.* We can apply Lemma B.6 to our problem for bounding $\text{TV}(p_y, \tilde{p}_y)$ from equation (6) and (10). If the condition in Lemma B.6 is satisfied, then

$$\text{TV}^2(p_y, \tilde{p}_y) \le \int_0^T \mathbb{E}\left[\|\nabla_{\boldsymbol{x}} \log \tilde{h}_y(t, \boldsymbol{y}_t) - \nabla_{\boldsymbol{x}} \log h_y(t, \boldsymbol{y}_t)\|^2 dt\right].$$

So the main goal is to estimate the bound of I (note that the boundedness of I is also the condition in Lemma B.6 we need)

$$\text{I} = \int_0^T \mathbb{E}\left[\|\nabla_{\boldsymbol{x}} \log \tilde{h}_y(t, \boldsymbol{y}_t) - \nabla_{\boldsymbol{x}} \log h_y(t, \boldsymbol{y}_t)\|^2 dt\right]$$

$$\le \sum_{i=0}^{k-1} \int_{t_i}^{t_{i+1}} \mathbb{E}\left[\|\nabla_{\boldsymbol{x}} \log h_y(t_i, \boldsymbol{y}_t) - \nabla_{\boldsymbol{x}} \log h_y(t, \boldsymbol{y}_t)\|^2\right] dt.$$

A direct result of Assumption 4 is $\|\nabla_{\boldsymbol{x}} \log h_y(t, \boldsymbol{x}) - \nabla_{\boldsymbol{x}} \log h_y(s, \boldsymbol{x})\| \le A\|\boldsymbol{x}\|\,|t - s|$. Therefore,

$$\text{I} \le \sum_{i=0}^{k-1} \int_{t_i}^{t_{i+1}} \mathbb{E}\left[\|\nabla_{\boldsymbol{x}} \log h_y(t_i, \boldsymbol{y}_t) - \nabla_{\boldsymbol{x}} \log h_y(t, \boldsymbol{y}_t)\|^2\right] dt$$

$$\le A^2 \sup_{t\in[0,T]} \mathbb{E}\left[\|\boldsymbol{y}_t\|^2\right] \sum_{i=0}^{k-1} \int_{t_i}^{t_{i+1}} (t - t_i)^2 dt$$

$$= \frac{A^2}{3} \frac{T^3}{k^2} \sup_{t\in[0,T]} \mathbb{E}\left[\|\boldsymbol{y}_t\|^2\right].$$

Then the next mission is to estimate $\mathbb{E}\left[\|\boldsymbol{y}_t\|^2\right]$. Recall $\boldsymbol{y}_t$ satisfy the equation (6), so

$$\boldsymbol{y}_t = \boldsymbol{y}_T + \int_T^t \boldsymbol{y}_s + 2\nabla_{\boldsymbol{x}} \log p_s(\boldsymbol{y}_s \mid y)ds + \sqrt{2}(\bar{B}_t - \bar{B}_T).$$

And thus

$$\mathbb{E}\left[\|\boldsymbol{y}_t\|^2\right] \leq \mathbb{E}\left[\|\boldsymbol{y}_T\|^2\right] + (T-t)\int_T^t \mathbb{E}\left[\|\boldsymbol{y}_s\|^2\right] ds$$

$$+ 4(T-t)\int_T^t \mathbb{E}\left[\|\nabla_{\boldsymbol{x}} \log p_s(\boldsymbol{y}_s \mid y)\|^2\right] ds + 2d(T-t)$$

$$\leq \mathbb{E}\left[\|\boldsymbol{y}_T\|^2\right] + T\int_T^t \mathbb{E}\left[\|\boldsymbol{y}_s\|^2\right] ds + 4LdT^2 + 2dT.$$

by the fact $\|\int_a^b F(x)dx\|^2 \leq (b-a)\int_a^b \|F(x)\|^2 dx$ and the following Lemma B.7. By setting $u(t) = \mathbb{E}\left[\|\boldsymbol{y}_t\|^2\right]$, $\lambda(t) = 4LdT^2 + 2dT$ and $\mu(t) = T$ in Grönwall's Inequality (Lemma B.8), we have

$$\sup_{t \in [0,T]} \mathbb{E}\left[\|\boldsymbol{y}_t\|^2\right] \leq (4LdT^2 + 2dT)e^{T^2} + \mathfrak{m}_2^2.$$

and therefore we get our final result,

$$\mathsf{TV}^2(p_y, \tilde{p}_y) \leq \frac{A^2}{3}T^3\left((4LdT^2 + 2dT)e^{T^2} + \mathfrak{m}_2^2\right) \cdot \frac{1}{k^2}. \qquad \square$$

**Lemma B.7** (Chen et al. (2023c))**.** *For any probability density function $p$ on $\mathbb{R}^d$, if $\log p$ is $L$-smooth, i.e. $\|\nabla^2 \log p\|_{\mathsf{op}} \leq L$, then*

$$\mathbb{E}^p\left[\|\nabla \log p\|^2\right] \leq Ld.$$

*Proof.* First, because $\log p$ is $L$-smooth,

$$|\Delta \log p| = \left|\mathrm{tr}(\nabla^2 \log p)\right| \leq Ld.$$

Then by the divergence theorem, we have $\int_{\mathbb{R}^d} \langle \nabla f, \nabla g \rangle \, dx = -\int_{\mathbb{R}^d} f\Delta g dx$ for any $f, g \in C^2(\mathbb{R}^d)$. Therefore,

$$\mathbb{E}^p\left[\|\nabla \log p\|^2\right] = \int_{\mathbb{R}^d} \langle \nabla \log p, \nabla \log p \rangle p dx = \int_{\mathbb{R}^d} \langle \nabla \log p, \nabla p \rangle \, dx$$

$$= -\int_{\mathbb{R}^d} p\Delta \log p dx \leq Ld. \qquad \square$$

**Lemma B.8** (Grönwall's Inequality)**.** *Let $u(t), \lambda(t), \mu(t) \in C([a,b])$. If $\mu(t) \geq 0$ for all $t \in [a,b]$ and*

$$u(t) \leq \lambda(t) + \int_a^t \mu(s)u(s)ds,$$

*then we have*

$$u(t) \leq \lambda(t) + \int_a^t \lambda(s)\mu(s) \exp\left(\int_s^t \mu(\tau)d\tau\right) ds.$$

*In particular, if $\lambda(t)$ is nondecreasing, then*

$$u(t) \leq \lambda(t) \exp\left(\int_a^t \mu(s)ds\right).$$

## C   IMPLEMENTATION DETAILS

### C.1   SETTING OF THE EXPERIMENT ON SYNTHETIC DATASET

For the synthetic data experiments, we train a multilayer perceptron (MLP) to model the score function at each timestep, $\nabla_{\boldsymbol{x}} \log p_{t_i}(\boldsymbol{x})$. We do not train a neural network classifier, since we adopt a simple Gaussian distribution as guidance, $\nabla_{\boldsymbol{x}} \log p_t(y \mid \boldsymbol{x}) = \mathcal{N}(\boldsymbol{x}; \boldsymbol{\mu}, \Sigma)$, whose gradient is tractable and can be computed analytically. Furthermore, the guidance parameter $\mathcal{N}(\boldsymbol{x}; \boldsymbol{\mu}, \Sigma)$ is kept time-invariant across all synthetic experiments.

## C.2 PRETRAINED MODELS AND REFERENCE SET

For the ImageNet-1K experiments, we adopt the unconditional diffusion model and classifiers provided by OpenAI, pretrained on ImageNet-1K at a resolution of $256 \times 256$, as well as conditional diffusion models at resolutions of $64 \times 64$, $128 \times 128$, and $256 \times 256$. To evaluate our generated samples, we compute FID, sFID, recall, and precision using reference batches of $10,000$ real images from ImageNet-1K, also provided by OpenAI.

## C.3 HYPERPARAMETERS

We first conducted experiments on synthetic data using a single NVIDIA RTX 4090 GPU for model training and sample generation. For real-world experiments, we employed four NVIDIA Tesla A100 GPUs (40GB) to generate samples from the ImageNet-1K dataset for quantitative evaluation. The hyperparameters for model training and sample generation are summarized in Table 3, Table 4, and Table 5.

Table 3: The hyperparameter settings of guided diffusion of synthetic data.

| Config | Value |
|---|---|
| training samples | 30000 |
| generated samples | 2000 |
| diffusion timesteps | 1000 |
| timestep respace | 250 |
| noise scheduler | cosine |
| optimizer | Adam |
| learning rate | 0.001 |
| training epoch | 2000 |
| classifier scale | 10.0 |
| batch size per GPU | 1024 |

Table 4: The training and generation hyperparameter settings of guided diffusion of CIFAR10.

| Config | Value |
|---|---|
| training samples | 50000 |
| training iterations of 1k classifiers | 100,000 |
| training iterations of 10 classifiers | 30,000 |
| batchsize | 64 |
| diffusion timesteps | 1000 |
| timestep respace | 250 |
| noise scheduler | cosine |
| optimizer | Adam |
| learning rate | 0.001 |
| classifier scale | 10.0 |

# D EXPERIMENTS FOR THE CONTRACTIVE PROPERTY

## D.1 TEST CONTRACTIVE PROPERTY ON TOY DATASETS

We set the synthetic datasets as same as the one in Section 4.1, that is, the target distribution $\boldsymbol{x}_0 \sim \mathcal{N}(\boldsymbol{x}_0; \boldsymbol{\mu}_0, \Sigma)$ with two classes $\{y_1, y_2\}$ and the corresponding classifiers are set by $p_0(y = y_1|\boldsymbol{x}) = \mathcal{N}(\boldsymbol{x}; \boldsymbol{\mu}_1, \Sigma)$ and $p_0(y = y_2|\boldsymbol{x}) = \mathcal{N}(\boldsymbol{x}; \boldsymbol{\mu}_2, \Sigma)$, where $\Sigma = I$, $\boldsymbol{\mu}_0 = (-6, 0)$, $\boldsymbol{\mu}_1 = (0, 6)$, and $\boldsymbol{\mu}_2 = (0, -6)$. So the conditional distributions are

$$p_0(\boldsymbol{x}|y = y_1) = \mathcal{N}\left(\boldsymbol{x}; \frac{\boldsymbol{\mu}_0 + \boldsymbol{\mu}_1}{2}, \frac{\Sigma}{2}\right), \quad p_0(\boldsymbol{x}|y = y_2) = \mathcal{N}\left(\boldsymbol{x}; \frac{\boldsymbol{\mu}_0 + \boldsymbol{\mu}_2}{2}, \frac{\Sigma}{2}\right).$$

Table 5: The hyperparameter settings of guided diffusion of ImageNet-1k.

| Config | Value |
| --- | --- |
| generated samples | 50000 |
| reference samples | 10000 |
| diffusion timesteps | 1000 |
| noise scheduler | cosine |
| attention resolutions | 32,16,8 |
| batch size per GPU | 4 |
| learn sigma | true |
| guidance scale | 1.0 |
| the number of fully connected layers | 4 |
| the number of hidden dimensions | 128 |
| use scale shift norm | true |

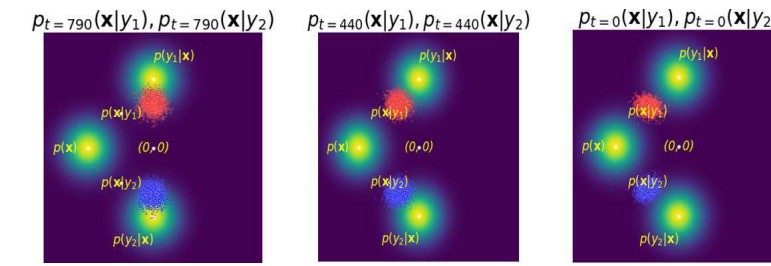

Figure 7: The reverse process which initially samples from the distribution $\mathcal{N}(\boldsymbol{x}; 0, I)$.

Under these settings, because the classifiers are Gaussian-like, i.e.

$$h_{y_1}(\boldsymbol{x}) = p_0(y = y_1|\boldsymbol{x}) \propto \exp(-\|\boldsymbol{x} - \boldsymbol{\mu}_1\|^2),$$
$$h_{y_2}(\boldsymbol{x}) = p_0(y = y_2|\boldsymbol{x}) \propto \exp(-\|\boldsymbol{x} - \boldsymbol{\mu}_2\|^2),$$

$\nabla_{\boldsymbol{x}}^2 \log h_{y_i} \equiv -I$, i.e. $h_{y_i}$ is 1-strongly log-concave. But it still cannot guarantee the contractive inequality in Theorem B.1. However, we can introduce a suitable guidance scale $s$ as mentioned in Remark 3 to make the contractive inequality valid.

To evaluate the contractive property, we consider two different initial sampling strategies.

First, we draw $\boldsymbol{x}_T$ from the standard Gaussian distribution $\mathcal{N}(\boldsymbol{x}_T; 0, I)$ to generate $p_0(\boldsymbol{x} \mid y_1)$ and $p_0(\boldsymbol{x} \mid y_2)$ with different guidance $p_0(y = y_1|\boldsymbol{x})$ and $p_0(y = y_2|\boldsymbol{x})$, respectively. The results are shown in Figure 7, where the final plot illustrates that the generated samples converge to the desired distribution. Comparing this result with Figure 2, where the initial sampling is $\mathcal{N}(\boldsymbol{x}; 0, I)p_0(y \mid \boldsymbol{x})$ to satisfy the condition in Theorem 3.1, we observe that sampling directly from $\mathcal{N}(\boldsymbol{x}; 0, I)$ still leads to the target conditional distribution due to the contractive property of (2).

Second, we initialize sampling from two arbitrary Gaussian distributions to further verify the contractive property. The results, shown in Figure 8, demonstrate that even with arbitrary Gaussian initializations, the contractive property of (2) ensures that the final generation converges to the target conditional distribution.

### D.2 TEST CONTRACTIVE PROPERTY ON THE IMAGENET-1K

We shift the initial sampling by adding a bias to the mean of the Gaussian distribution, resulting in samples from $\mathcal{N}(\boldsymbol{\mu}, I)$. To evaluate class-conditional generation, we compare an unconditional diffusion model with classifier guidance against a conditional diffusion model without guidance, as shown in Figure 9. With a bias of $\boldsymbol{\mu} = 0.03 \cdot \mathbb{1}$ or $\boldsymbol{\mu} = -0.03 \cdot \mathbb{1}$ added to the standard Gaussian distribution, the classifier-guided diffusion model continues to generate class-consistent images. In contrast, the conditional diffusion model without guidance produces images that are

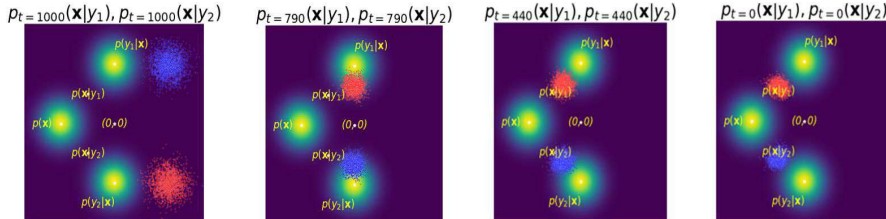

Figure 8: This initially samples from two arbitrary Gaussian distributions, both reverse processes can reconstruct the $p_0(\boldsymbol{x} \mid y_i)$ under the guidance of $p_0(y_i \mid \boldsymbol{x})$.

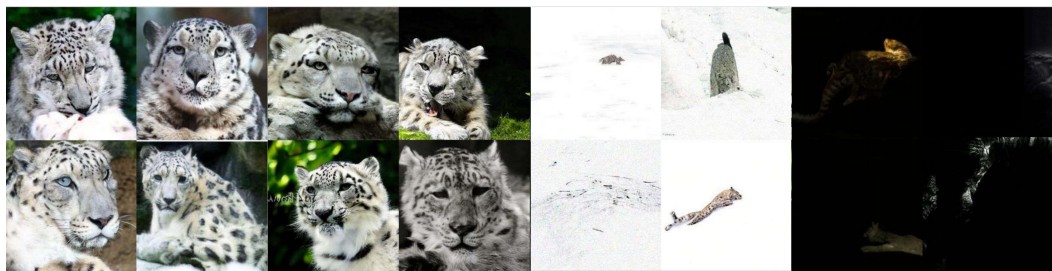

Figure 9: The images are sampled by adding positive bias $\mathcal{N}(\boldsymbol{x}; \boldsymbol{0}, +\boldsymbol{\mu})$ and negative bias $\mathcal{N}(\boldsymbol{x}; \boldsymbol{0}, -\boldsymbol{\mu})$ onto the final distribution $P_T(\boldsymbol{x})$. The left 8 images are generated using an unconditional diffusion model with classifier guidance at eight timesteps, and the left 8 failed samples are generated using a conditional diffusion model without guidance. Class: snow leopard
.

either overly bright or overly dark, both of poor quality. These results demonstrate that classifier guidance improves the robustness of diffusion models to shifts in the initial distribution.

In addition, by setting the positive bias $\boldsymbol{\mu}+ = 0.015 \cdot \mathbb{1}$ and the negative bias $\boldsymbol{\mu}- = -0.03 \cdot \mathbb{1}$, we generate samples using an unconditional diffusion model guided by classifiers corresponding to 8 timesteps. In each subfigure, the classifier guidance scale increases from left to right in the order 0, 1, 2.5, 5, 7.5, 10, as shown in Figures 10, 11, and 12. The bright samples correspond to the positive bias $\boldsymbol{\mu}+$, while the dark samples correspond to the negative bias $\boldsymbol{\mu}-$. As illustrated by the generated results, sample quality consistently improves as the classifier scale increases.

# E  MORE EXPERIMENTS ON IMAGENET-1K

The Figure 13 shows the samples generated with an unconditional diffusion model guided by classifiers $p_t(y|\boldsymbol{x})$ corresponding to 8 timesteps: $t = 875, 750, 625, 500, 375, 250, 125, 0$.

# F  USE OF LARGE LANGUAGE MODELS

For writing this manuscript, we used OpenAI's GPT-5 (ChatGPT) solely for language polishing and minor stylistic improvements. All technical content, results, derivations, and experiments were developed independently by the authors. No scientific content, data, proofs, or results were generated or altered by the model.

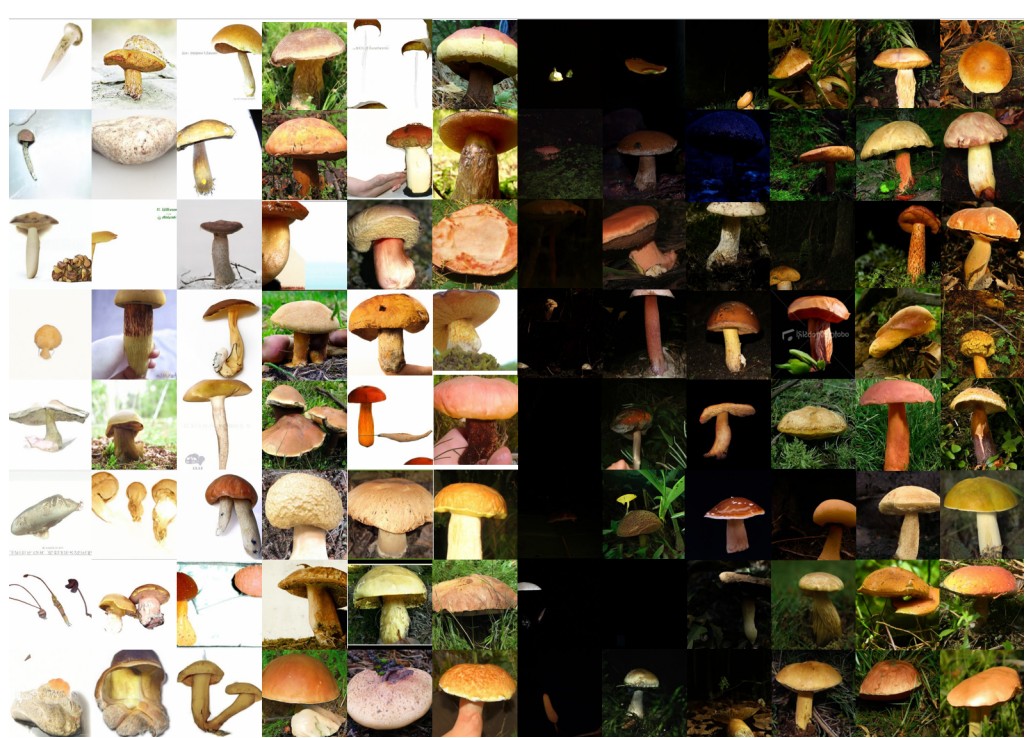

Figure 10: Adding negative bias(left) and positive bias(right) on the initial sampling. In each row, for each of the six left images (positive bias) and the six right images (negative bias), the classifier scale corresponds to (0, 1.0, 2.5, 5.0, 7.5, 10). The class is 997: bolete.

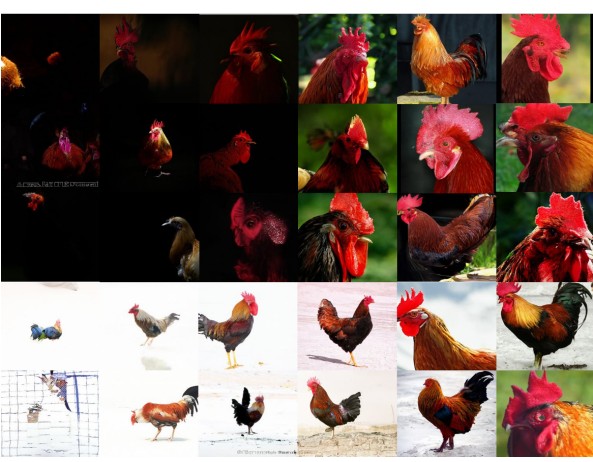

Figure 11: The top 3 rows correspond to the result of adding negative and the bottom 2 rows correspond to positive bias on the initial sampling. For six images in each row, the classifier scale gradually increases from $0.0$ to $10.0$ from left to right, class is 7: cock.

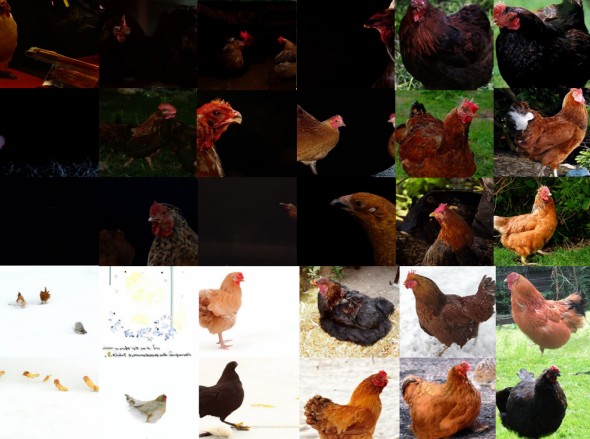

Figure 12: The result of adding negative and positive bias on the initial sampling. In each row, the classifier scale gradually increases from $0.0$ to $10.0$ from left to right, class is 8: hen.

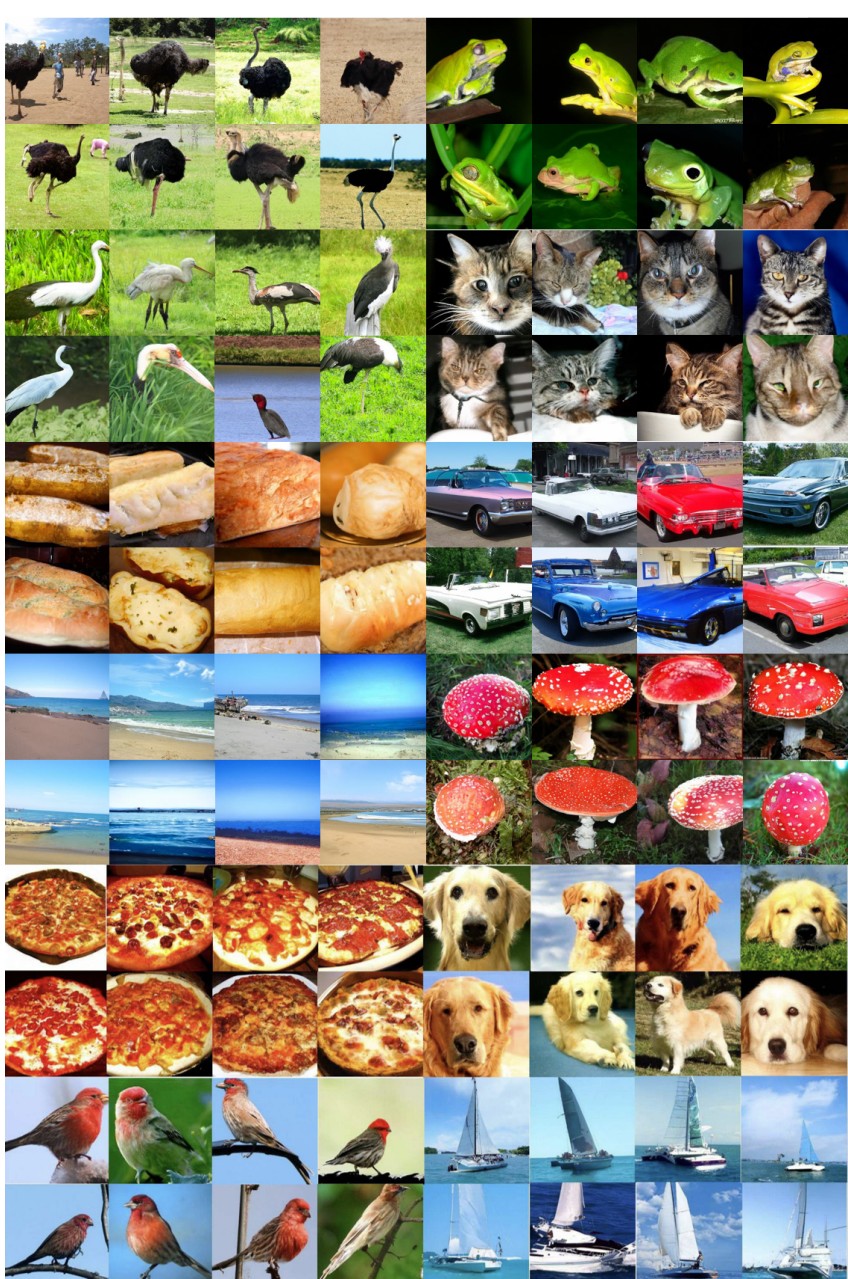

Figure 13: Generated samples guided by classifiers correspond to eight timesteps (FID: 12.90). Classes are 9: ostrich, 31: tree frog, 134: crane, 281: tabby cat, 930: French loaf, 511: check, 978: seashore, 992: agaric, 963: pizza, 207: golden retriever, 15: robin, 484: catamaran.

