# OpenReview forum: "Analyzing Time-independent Classifiers for Conditional Generation"
_ICLR.cc/2026/Conference — Submitted to ICLR 2026_

### Official Review · Reviewer_9hyn · 2025-10-29

**Soundness:** 3
**Presentation:** 3
**Contribution:** 2
**Rating:** 4
**Confidence:** 4

**Summary:**

This paper investigates the necessity of time-dependent classifiers in classifier guidance for conditional generation with diffusion models. The authors theoretically and empirically show that, under certain conditions, guiding diffusion sampling with a single time-independent classifier, or a small number of such classifiers trained on data at different noise levels, can achieve sample quality comparable to the standard approach that uses a full time-dependent classifier. The work provides theoretical convergence guarantees, proposes a refined multi-classifier approach practical for real-world data manifolds, and backs claims with experiments on both synthetic low-dimensional data and large-scale image datasets.

**Strengths:**

- **Strong Theoretical Analysis:** The paper offers thorough mathematical treatment of both the feasibility and limitations of time-independent classifier guidance, including explicit proofs. Notably, the contractive property of the reverse process with a strongly log-concave classifier is both formally and intuitively described.
- **Algorithmic Innovation:** The proposed strategy of using a small set of time-independent classifiers addresses a significant computational bottleneck in existing classifier guidance methods, with the transition dynamics rigorously constructed and justified.
- **Clear Empirical Evaluation:** Comprehensive and comparative experiments are conducted on both toy and real-world datasets. Figure 3 demonstrates clear visual improvements in sample quality as the number of classifiers increases, and Table 1  provides direct quantitative support: even 8 classifiers approach the FID/sFID of 1000-classifier guidance.
- **Relevance and Impact:** The work provides actionable insight for improving the efficiency and scalability of classifier-guided diffusion models, which is a topic of considerable interest in generative modeling.
- **Clarity in Exposition:** The paper is well structured with a clear flow from theoretical motivation, through algorithmic proposal, to empirical validation. Mathematical notation is generally consistent, and practical implications are repeatedly emphasized.

**Weaknesses:**

1. **Potential Overstatement on Real-World Applicability of Single Classifier Guidance:** Theoretically, the feasibility of using a single time-independent classifier depends critically on strong log-concavity and the ability to sample from $\mathcal{N}(\mathbf{x}; 0, I)p_0(y \mid \mathbf{x})$, which is not realistic for high-dimensional natural images. Although the authors acknowledge this, empirical validation on real-world data is only performed for the multi-classifier regime. There is a substantial gap between theory and practice unaddressed for the single-classifier setting.
2. **Missing or Limited Comparison to Related Alternative Guidance Approaches:** While classifier-free guidance is briefly mentioned, there is insufficient experimental or conceptual comparison, especially given classifier-free guidance’s considerable practical importance. For instance, Table 1 and Table 2 compare only varying counts of time-dependent classifiers and lack strong baselines from other guidance paradigms.
3. **Empirical Evidence for Robustness to Classifier and Data Bias:** Experiments such as those visualized in Figures 9, 10, and 11 focus on varying guidance strength and initial distribution bias, but they predominantly demonstrate results for a handful of selected classes. The breadth and depth of robustness testing are not thoroughly quantified.
4. **Limited Discussion of Limitations and Potential Failure Modes:** The discussion lightly touches on the failure of single-classifier guidance for manifold data, but does not explore or visualize where/why this approach might entirely fail for more complex datasets or tasks (e.g., text-to-image, compositionality, or classes with high inter-class visual similarity). Similarly, sensitivity to classifier architecture and capacity is not explored.
5. **Insufficient Empirical Comparison with Robust/Adversarial Guidance:** Recent works such as Kawar et al. (2023), which explore the robustness of classifier-guided diffusion via adversarial training, are not discussed or compared in depth. Such comparisons could reveal whether the proposed approach has complementary or inferior/robustness properties.

***Reference***
- Kawar B, Ganz R, Elad M. Enhancing diffusion-based image synthesis with robust classifier guidance. arXiv preprint arXiv:2208.08664. 2022 Aug 18.

**Questions:**

- How does the approach perform when the noise level/timestep selection for classifier training is heavily skewed (e.g., all classifiers from early or late timesteps)? Is uniform selection optimal, or might adaptive selection perform better?
- Can the authors more rigorously quantify the robustness of the method to misspecification of the classifier(s) or class imbalance, both theoretically and empirically?
- Is there any scenario (e.g., compositional generation, text-to-image tasks) where the saturation with 8-16 classifiers no longer holds? Can the approach handle such settings, or is it fundamentally limited to simple class-conditional tasks?
- How would the method compare, practically and in terms of efficiency/quality, to classifier-free guidance or adversarially robust classifier guidance approaches? A small ablation or discussion on this point could substantially strengthen the story.
- What is the computational overhead or memory reduction in practical training settings (e.g., on larger-scale or resource-constrained problems) for various values of $k$? Is there a practical lower bound for $k$ where gains taper off or instability emerges?

---

> ### Author Response · Authors · 2025-11-21
>
> We sincerely thank the reviewer for the detailed and thoughtful feedback. We respond to each point below.
>
> **1. Limitations of single-classifier guidance**
>
> For Weakness 1, we apologize that our presentation did not make this point sufficiently clear. We have already evaluated the case of using a single time-independent classifier. As shown in Figure 1, the left panel reports a confidence score of 0 for single-classifier guidance, and the right panel (first column) shows samples generated using only one classifier. These results demonstrate that relying on a single classifier is not realistic for high-dimensional real-world data. This failure aligns with the manifold hypothesis discussed in Section 3.4, which explains why a single classifier cannot capture the structure of complex data manifolds.
>
> **2. Comparison with classifier-free guidance (CFG)**
>
> For Weakness 2 and Question 4, our method can be directly applied to classifier-free guidance (CFG), since CFG and classifier guidance (CG) are theoretically equivalent in their objective of estimating the guidance term $\nabla_{x} \log p_t(y \mid x)$. In CG, a time-dependent classifier is trained to approximate this quantity. In CFG, the network $s_\theta(t, x, y)$ estimates the conditional score $\nabla_{x} \log p_t(x \mid y)$, and $s_\theta(t, x, \emptyset)$ estimates the unconditional score $\nabla_{x} \log p_t(x)$. Their difference $s_\theta(t, x, y) - s_\theta(t, x, \emptyset)$ thus approximates $\nabla_{x} \log p_t(y \mid x)$. The core objective of CFG is therefore also to train the time-dependent estimator $s_\theta(t, x, y)$.
>
> Based on our theoretical analysis, this estimator can similarly be replaced by $k$ time-independent networks $s_{\theta_1}(x_{t_1}, y), \ldots, s_{\theta_k}(x_{t_k}, y)$, followed by the corresponding reverse process. We acknowledge that the experimental comparison with CFG is not included in the current version. These experiments are ongoing, and we will report both the results and the efficiency comparison between our CFG variant and the standard CFG formulation once they are ready.
>
> **3. Robustness evaluation and lack of quantitative metrics**
>
> For Weakness 3, we apologize for not providing quantitative robustness results. Figures 9, 10, and 11 were intended to visualize the contractive properties of the reverse process described in Theorem 3.3. They show that even for real-world datasets—where the assumptions in Theorem 3.3 may not strictly hold—the reverse process still exhibits contractive behavior and is robust to variations in the initial sampling distribution. The details are provided in Section 4.3 and Appendix B.1. In the revised version, we will additionally include quantitative measurements of how the generated sample quality changes with respect to variations in the guidance scale.
>
> **4. Applicability to more complex datasets and tasks**
>
> For the first part of Weakness 4 and Question 3, we acknowledge that we have not yet tested our method on more complex tasks such as compositional generation or text-to-image generation. At this stage, our primary objective is to develop a theoretical understanding of the proposed model. The experiments on image generation were intended as initial validation of the theoretical findings. Exploring more complex datasets and tasks is an important next step, and we will pursue this direction in future work.

---

> > ### Author Response · Authors · 2025-11-21
> >
> > **5. Sensitivity to classifier architecture and adversarial robustness**
> >
> > For the second part of Weakness 4, Weakness 5, and Question 2, we acknowledge that sensitivity analysis with respect to classifier structure, training algorithm, and adversarial robustness lies outside the current scope of our study. These topics are indeed valuable and relevant, and we thank the reviewer for highlighting them. Our present focus is on examining the feasibility of using time-independent classifiers for conditional generation. We will consider robustness-oriented comparisons—including adversarially trained classifiers—in future work.
> >
> > **6. Effects of timestep selection**
> >
> > For Question 1, as analyzed in Section 3.4, the selected classifiers must collectively recover the manifold structure. This requires (i) classifiers trained on noisy data from early diffusion steps to provide meaningful guidance far from the manifold, and (ii) classifiers trained on data near the manifold to guide the final refinement. This motivates the strategy of selecting 10 timesteps uniformly from all 1000 steps. We apologize for not providing empirical evidence of this design choice. We are currently running two additional experiments, inspired by the reviewer’s suggestion: selecting 10 timesteps only from the first 100 steps, and selecting 10 timesteps only from the last 100 steps. Due to limited computational resources, we are waiting for the results.
> >
> > **7. Practical selection of $k$**
> >
> > For Question 5, we acknowledge that we provide only a simple theoretical analysis regarding the choice of $k$, namely that the upper bound scales as $\mathcal{O}(1/k)$. Figure 1 practically showed how $k$ influences confidence scores. However, since our experiments focus on image generation, we have not yet thoroughly explored the practical selection of $k$ for other tasks or more complex datasets. We plan to investigate the effect of $k$ more comprehensively across tasks and architectures in future work.

---

### Official Review · Reviewer_n8i4 · 2025-10-31

**Soundness:** 3
**Presentation:** 3
**Contribution:** 2
**Rating:** 4
**Confidence:** 4

**Summary:**

This paper tackles the high computational cost of classifier-guided diffusion models (CGDM). Traditional CGDM requires a classifier at every diffusion step. The authors argue this dense guidance is unnecessary.
They propose a more efficient method. Their approach uses a small set (k) of time-independent classifiers. Each classifier is trained on data from only a few selected timesteps. The authors provide a theoretical bound showing the error scales as O(1/k).
Experiments on ImageNet-1K show the sparse approach (e.g., k=8) achieves similar sample quality to the traditional method.

**Strengths:**

The paper is well-organized and provides sufficient theoretical justification. The authors begin by discussing the ideal case of a single classifier and highlight its limitations, which naturally leads to the proposed solution of using multiple classifiers.
The technical details, including the introduction of a new transition probability, the analysis of the contractive property, and the O(1/k) convergence bound, are presented clearly, offering strong support for the method within the CGDM framework. Additionally, the experimental results on ImageNet strongly support the central claim, showing that significant efficiency improvements can be achieved by reducing the number of classifiers from T to k without a notable loss in performance.

**Weaknesses:**

1.A key weakness is that the paper's contribution is narrowly focused on the traditional CGDM. This is a limitation because the research community has largely shifted towards Classifier-Free Guidance (CFG). CFG was introduced specifically to avoid the high cost of external classifiers—the very problem this paper attempts to mitigate. Consequently, the paper primarily addresses a challenge that the current, dominant paradigm was designed to circumvent entirely, which may limit the practical significance of its contribution.

2.the paper lacks an experimental comparison with a CFG baseline. While it demonstrates that sparse CGDM is more efficient than dense CGDM, it does not compare it to the widely used CFG method. Without such a comparison, it is difficult to assess the practicality and competitiveness of the proposed approach.

3.In Table 2, the k=10 model for CIFAR-10 (7.36 FID) significantly outperforms the k=1000 baseline (19.36 FID), while also using fewer training iterations (30k vs. 100k). This result seems counter-intuitive and contradicts the performance observed on ImageNet, suggesting that the baseline model may not have been adequately trained, thus weakening the credibility of the paper's experimental results.

**Questions:**

1.Given the widespread use of CFG in the generation field, I encourage the authors to elaborate on the motivation for optimizing the CGDM framework. Are there specific scenarios where sparse CGDM offers a distinct advantage?

2.A quantitative comparison with a CFG-based baseline would help demonstrate the robustness of the proposed method in practical scenarios.

3.Additional clarification on the CIFAR-10 results in Table 2 would be helpful. Could the authors explain the large performance gap between the k=10 and k=1000 models? Considering the differing number of training iterations, would it be necessary to further discuss the fairness of the baseline?

---

> ### Author Response · Authors · 2025-11-21
>
> We sincerely thank the reviewer for the thoughtful and constructive feedback. We address each point in detail below.
>
> **1. Comparison with classifier-free guidance (CFG)**
>
> For Weakness 1, Weakness 2, Question 1, and Question 2 related to CFG, we emphasize that our idea can be directly applied to classifier-free guidance because CFG and classifier guidance (CG) are theoretically equivalent. Both methods aim to estimate the guidance term $\nabla_{x} \log p_t(y \mid x)$. In CG, this term is approximated by a time-dependent classifier. In CFG, the network $s_\theta(t, x, y)$ estimates the conditional score $\nabla_{x} \log p_t(x \mid y)$, and $s_\theta(t, x, \emptyset)$ estimates the unconditional score $\nabla_{x} \log p_t(x)$. The difference $s_\theta(t, x, y) - s_\theta(t, x, \emptyset)$ therefore approximates $\nabla_{x} \log p_t(y \mid x)$.
>
> Since CFG also trains a time-dependent estimator $s_\theta(t, x, y)$, our theoretical analysis indicates that one can replace it with $k$ time-independent networks $s_{\theta_1}(x_{t_1}, y), \ldots, s_{\theta_k}(x_{t_k}, y)$ and perform the corresponding reverse process for generation. We acknowledge that we did not include CFG experiments in the submission. These experiments are ongoing, and we will report the results, along with a comparison of efficiency between our proposed CFG formulation and the standard CFG method, as soon as they are available.
>
> **2. Clarification regarding CIFAR-10 performance**
>
> For Weakness 3 and Question 3, we appreciate the reviewer’s careful observation regarding the CIFAR-10 results in Table 2, where the $k=10$ model (7.36 FID) appears to outperform the $k=1000$ baseline (19.36 FID). This discrepancy is an artifact of evaluating the baseline under a fixed, suboptimal guidance scale of $s = 7.5$. After conducting an ablation study with CFG scales ranging from $0$ to $10$, we found that the $k=1000$ baseline achieves its optimal performance at $s = 1.0$. Under this optimal setting, the baseline achieves FID$ = 4.06$, which is significantly better than the FID of the $k=10$ model. This confirms that the baseline was fully trained and that the previously reported result was due solely to the guidance scale rather than inadequate training. We are currently determining the optimal guidance scale for the $k=10$ model and will update Table 2 so that all models are reported under their optimal settings. This will resolve the apparent inconsistency and ensure a fair comparison.
>
> Baseline ($k = 1000$) performance at optimal guidance scale ($s = 1.0$):
>
> FID = $4.06$, sFID = $4.22$, IS = $5.54$, Precision = $0.6668$, Recall = $0.5822$.

---

### Official Review · Reviewer_RjtS · 2025-11-01

**Soundness:** 2
**Presentation:** 3
**Contribution:** 2
**Rating:** 4
**Confidence:** 3

**Summary:**

This paper was analyzing time-independent classifiers for conditional generation and proposed interesting methods. However, several aspects of the paper could be strengthened, particularly in terms of experimental completeness and comparison with classifier-free guidance methods.

**Strengths:**

The paper presents an interesting and theoretically grounded approach to improve the efficiency of classifier-guided diffusion models by replacing time-dependent classifiers with a small set of time-independent classifiers.
The authors provide solid theoretical analysis, including the feasibility of using a single time-independent classifier trained on clean data and the bounds on convergence. This theoretical framework adds credibility to the proposed approach.

**Weaknesses:**

The experiment is insufficient:
1. While the paper discusses classifier-free guidance in the related work, no experimental comparison is provided. Since classifier-free guidance is now a widely adopted alternative that also aims to reduce the dependency on explicit classifiers, it is crucial to evaluate the proposed method against it in terms of generation quality, computational efficiency, and scalability.
2. The paper claims that a small number of classifiers trained at selected timesteps suffice for high-quality generation. However, the criteria for selecting these timesteps and their impact on performance are not discussed.
3. Relying solely on quantitative metrics such as FID and sFID may not fully capture perceptual quality. Incorporating additional perceptual metrics (e.g., LPIPS, CLIPScore) and a user study would provide more holistic evidence of visual realism and human-perceived quality.
4. The paper repeatedly emphasizes computational efficiency, yet no explicit runtime or FLOPs analysis is provided. It would be highly beneficial to include a comparison table. eg.Training and inference time for different numbers of classifiers and memory and computational cost.
5. The theoretical results rely on the assumption that the classifier is strongly log-concave. The authors themselves acknowledge (L267–269) that this assumption does not hold in practice for complex neural classifiers. Hence, more empirical evidence on larger, real-world datasets is needed to show that the proposed method still performs robustly even when classifiers are non-convex.

**Questions:**

Refer to the Weakness.

---

> ### Author Response · Authors · 2025-11-21
>
> We sincerely thank the reviewer for the thoughtful and constructive feedback. We appreciate the time invested in assessing our work and the suggestions that help us strengthen both the theoretical discussion and the experimental evaluation. Below, we address each point in detail.
>
> **1. Comparison with classifier-free guidance (CFG)**
>
> We appreciate the reviewer’s suggestion to include experimental comparisons with classifier-free guidance. Our framework can indeed be directly applied to the CFG setting because CFG and classifier-guided (CG) approaches share the same theoretical objective of estimating the guidance term $\nabla_{x} \log p_t(y \mid x)$. In CG, this quantity is approximated by a time-dependent classifier. In CFG, the difference $ s_\theta(t, x, y) - s_\theta(t, x, \emptyset)$ serves as an estimator of the same guidance term.
>
> Building on our theoretical analysis, the time-dependent conditional network in CFG can similarly be replaced by a small collection of time-independent estimators $s_{\theta_1}(x_{t_1},y),\cdots,s_{\theta_k}(x_{t_k},y)$ trained at selected timesteps, followed by the corresponding reverse diffusion process. We acknowledge that we did not include CFG experiments in the initial submission. These experiments are currently running, and we will report both the generation results and the efficiency comparison with standard CFG once they are completed.
>
> **2. Criteria for selecting timesteps**
>
> As discussed in Section 3.4, the subset of timesteps must satisfy two requirements: (i) the corresponding classifiers must provide meaningful guidance for highly noisy samples at the early stages of the reverse process, and (ii) they must also effectively guide samples around the target data manifold at the late stages of reverse process. This dual role implies that the selected timesteps should uniformly span the entire diffusion trajectory.
>
> Our current choice of selecting 10 uniformly spaced timesteps from the full 1000-step schedule is motivated by this consideration. We agree that empirical evidence supporting this strategy is valuable. To investigate this further, we are conducting two additional ablation experiments: selecting 10 timesteps exclusively from the first 100 steps, and selecting 10 timesteps exclusively from the last 100 steps. Due to limited computational resources, these experiments are still ongoing, but we will include their results in the revision.
>
> **3. Additional perceptual metrics**
>
> We thank the reviewer for recommending perceptual metrics such as LPIPS or CLIPScore. However, as our task is unconditional image generation, the appropriate evaluation focuses on distribution-level discrepancies between generated and real samples. CLIPScore is primarily designed for text-to-image generation and measures semantic alignment with textual prompts. LPIPS evaluates perceptual similarity between paired images, making it more suitable for reconstruction or editing tasks, where ground-truth reference images are available. Since unconditional image generation does not provide such paired data, these metrics are less informative. For this reason, we rely on FID and sFID, which directly assess the quality of the generated sample distribution.
>
> **4. Computational efficiency analysis**
>
> We appreciate the reviewer’s emphasis on computational analysis. We are currently finalizing a detailed comparison of convergence speed, measuring the training time required to reach a specified accuracy target for models using 10 versus 1000 classifiers. This experiment directly illustrates the primary computational benefit of our approach: substantial reductions in training cost and improved scalability.
>
> Regarding inference memory and FLOPs, we respectfully clarify that our work specifically addresses training-time efficiency rather than inference-time optimization. Because the inference phase is not the focus of our contribution, we do not include such analysis. In the revision, we will make this scope explicit to avoid misunderstanding.
>
> **5. Log-concavity assumption**
>
> We acknowledge the reviewer’s concern that the strong log-concavity assumption does not hold for modern neural classifiers. In our theoretical analysis, this assumption ensures that the initial sampling distribution can be replaced by a standard Gaussian, as discussed in Section 3.3. Despite this idealized condition, our experiments on larger real-world datasets such as ImageNet and CIFAR demonstrate that sampling directly from a standard Gaussian still yields high-quality results, even when we cannot guarantee  the convavity. Furthermore, empirical evidence presented in Section 4.3 and Appendix D.2 shows that the reverse diffusion process retains a contractive property in practice, supporting the robustness of our method beyond the theoretical assumptions.

---

### Author Response · Authors · 2025-12-03
**Final Response**

We thank all reviewers for their constructive feedback and valuable suggestions, which have helped us significantly improve the clarity and scope of the paper. In the revised version, we have added Section 4.4 to present the results of applying our idea to classifier-free guidance on the CIFAR-10 dataset. The visualization results in this new section support the validity of extending our approach to CFG and demonstrate that time-independent classifiers can provide meaningful guidance even in this widely used framework.

Due to the limited time available during the rebuttal period, as well as our constrained computational resources, we are unable to include additional experiments on ImageNet or to provide a fully fair comparison of training efficiency between our approach and standard CFG. We sincerely apologize for this limitation. Nevertheless, we believe the new results added during the rebuttal clearly strengthen the central message of our work and further validate the theoretical and empirical insights presented.

We thank the reviewers again for their time and thoughtful evaluations.

---

### Meta-Review · Area_Chair_gmiZ · 2026-01-05

**Summary:**

This paper proposes using a small set of time-independent classifiers to guide diffusion models instead of training a classifier for every timestep. Reviewers praised the strong theoretical analysis and the potential for better training efficiency. However, they all pointed out a major lack of comparison with Classifier-Free Guidance (CFG), which is the current standard. There were also concerns about unfair baselines and whether the method works on complex real-world data. Before the rebuttal, reviewers felt the experiments were incomplete without CFG results. During the discussion, the authors promised these experiments but could not finish them all in time. Therefore, I recommend rejection and encourage the authors to add these key comparisons for the next version.

**Reviewer Concerns:**

The rebuttal successfully clarified the theoretical connection to CFG and explained why the initial CIFAR-10 baseline looked weak. However, the most critical concern remains outstanding because the authors did not provide the actual experimental results comparing their method against standard CFG.

**Reviewer Scores:**

Reviewer RjtS would likely keep their low score because the requested runtime analysis and CFG comparisons are still missing. Reviewer n8i4 might have raised their score if the authors had successfully demonstrated that their method competes well with CFG. Reviewer 9hyn would likely maintain their score as they still have doubts about the method's robustness on complex tasks.

---

### Decision · Program_Chairs · 2026-01-26

Reject